# Modulated Diffusion:
# Accelerating Generative Modeling with Modulated Quantization

Weizhi Gao [1]  Zhichao Hou [1]  Junqi Yin [2]  Feiyi Wang [2]  Linyu Peng [3]  Xiaorui Liu [1]

## Abstract

Diffusion models have emerged as powerful generative models, but their high computational cost in iterative sampling remains a significant bottleneck. In this work, we present an in-depth and insightful study of state-of-the-art acceleration techniques for diffusion models, including caching and quantization, and reveal their limitations in computation error and generation quality. To break these limits, this work introduces Modulated Diffusion (MoDiff), an innovative, rigorous, and principled framework that accelerates generative modeling through modulated quantization and error compensation. MoDiff not only inherits the advantages of existing caching and quantization methods but also serves as a general framework to accelerate all diffusion models. The advantages of MoDiff are supported by solid theoretical insight and analysis. In addition, extensive experiments on CIFAR-10 and LSUN demonstrate that MoDiff significantly reduces activation quantization from 8 bits to 3 bits without performance degradation in post-training quantization (PTQ). Our code implementation is available at https://github.com/WeizhiGao/MoDiff.

## 1. Introduction

Diffusion models have emerged as powerful generative models for producing high-quality data samples, ranging from images to audio and beyond (Ho et al., 2020; Song et al., 2021a;b). These models work by iteratively transforming a simple noise distribution into complex, structured outputs, guided by a learned reverse diffusion process. Despite their effectiveness, diffusion models come with significant

computational costs (Liu et al., 2022; Li et al., 2023). The iterative nature of the sampling process, which requires multiple inferences through neural networks, makes these models computationally expensive and time-intensive. This limitation restricts their scalability and accessibility.

Existing work aims to enhance the efficiency of the sampling process in diffusion models through several strategies. Caching methods, for example, accelerate diffusion models by reusing intermediate computations (Ma et al., 2024b; Wimbauer et al., 2024). These methods exploit significant similarities between features at nearby time steps, enabling the skipping of redundant computations by directly using cached results. Additionally, quantization techniques reduce inference costs by converting model weights and activations into integers using scaling factors (Nagel et al., 2021; Yang et al., 2019). Among these, post-training quantization (PTQ) stands out as a promising approach since it estimates scaling factors in a training-free manner, making it broadly applicable to pre-trained models (Li et al., 2021). Another line of work focuses on efficient sampling strategies with solvers or samplers, such as denoising diffusion implicit models (DDIMs), which significantly reduce the number of sampling steps required in diffusion models, and speed up the process (Song et al., 2021a).

Our preliminary studies reveal that while caching and PTQ methods have achieved notable success in accelerating the sampling process, they also introduce significant limitations. First, our analysis reveals that caching methods can lead to reuse errors that accumulate throughout the generation process, particularly when reuse schedules are not carefully designed with respect to the time step and reused components. For instance, when following the reuse strategy of DeepCache (Ma et al., 2024b), but slightly modifying the reused components, we observe that the relative $\ell_2$ distance between the features of a standard diffusion model and those of caching methods increases significantly throughout the generation process, reaching 40% in the final step, even when the cache is updated every three steps. On the other hand, our studies show that diffusion models exhibit significant outliers in activations and variations in activation ranges across time steps, leading to substantial quantization errors under low-bit activation quantization.

[1]Department of Computer Science, North Carolina State University [2]National Center for Computational Science, Oak Ridge National Lab [3]Department of Mechanical Engineering, Keio University. Correspondence to: Xiaorui Liu <xliu96@ncsu.edu>.

*Proceedings of the 42nd International Conference on Machine Learning*, Vancouver, Canada. PMLR 267, 2025. Copyright 2025 by the author(s).

In this work, we propose **Modulated Diffusion (MoDiff)**, an innovative, rigorous, and principled framework that accelerates the diffusion sampling process while addressing the limitations of existing methods. Specifically, we propose modulated computation to significantly reduce activation quantization error by leveraging the computation redundancy across the diffusion time steps. Moreover, we further introduce a novel error compensation modulation to address error accumulation. Furthermore, we provide theoretical analyses to explain why the temporal difference results in lower quantization error and how error compensation effectively eliminates accumulated errors. Our extensive experiments validate the effectiveness of this framework, demonstrating that MoDiff pushes the activation quantization limit of PTQ methods from 8 bits to as low as 3 bits without any performance degradation, all within a training-free manner.

The proposed MoDiff framework inherits the advantages of existing acceleration methods while addressing their limitations. It significantly generalizes the caching techniques through modulated computation but reduces apprpoximation and accumulated error. Additionally, from a quantization perspective, MoDiff significantly reduces the quantization error of existing PTQ methods, enabling the use of much lower activation bit-widths without sacrificing performance. Notably, MoDiff is agnostic to quantization algorithms and can be generally applied across different methods, making its contribution orthogonal to existing PTQ techniques. Furthermore, MoDiff imposes no constraints on samplers, ensuring compatibility with solver-based acceleration methods. In summary, our main contributions are as follows:

- We present an in-depth and insightful preliminary study that reveals the limitations of existing acceleration techniques for diffusion models, such as caching and quantization methods, highlighting issues like error accumulation and high approximation error.

- We propose MoDiff, a novel, rigorous, and principled framework that accelerates diffusion models through modulated quantization and error compensation. MoDiff not only inherits the advantages of existing methods but also overcomes their limitations, enabling significantly more aggressive activation quantization.

- We provide theoretical analyses of quantization error and the error compensation mechanism in MoDiff, demonstrating that our approach can significantly reduce the required activation bit precision in PTQ.

- Extensive experiments on CIFAR-10, LSUN-Churches, and LSUN-Bedroom show that MoDiff enables state-of-the-art quantization techniques to reduce activation precision from 8 bits to as low as 3 bits without any performance degradation in a training-free manner.

## 2. Related Work

**Diffusion Models.** Diffusion models have become a cornerstone of generative modeling, achieving remarkable success across diverse domains such as image synthesis, data distillation, and molecular modeling (Ho et al., 2020; Hoogeboom et al., 2022; Su et al., 2024). These models operate on an iterative framework that involves adding noise in the forward process and learning to remove it during the reverse process (Dhariwal & Nichol, 2021). However, the iterative nature of the sampling process makes generation computationally expensive (Song et al., 2021b; Ho et al., 2020). To address this efficiency bottleneck, a line of research has focused on improving the sampling process by optimizing the variance schedule or employing more advanced ODE solvers (Song et al., 2021a; Nichol & Dhariwal, 2021; Liu et al., 2022). For example, Denoising Diffusion Implicit Models (DDIMs) introduce a non-Markovian formulation for the diffusion process, significantly reducing the number of sampling steps required (Song et al., 2021a).

**Caching Methods.** Caching methods for accelerating diffusion models aim to reduce redundant computations during the generative process by reusing intermediate results, thereby improving efficiency (Ma et al., 2024b; Wimbauer et al., 2024; Ma et al., 2024a). These strategies address the high computational cost by selectively storing intermediate states from the reverse diffusion process for reuse in subsequent steps. For example, DeepCache reuses cached upsampled features every $N$ time steps (Ma et al., 2024b). However, it can accumulate errors in the generation process with the reusing technique. Existing works rely on heuristic approaches to determine $N$, which limits its generalizability. Some methods also attempt to preserve model performance by fine-tuning diffusion models, but this approach can be computationally expensive (Wimbauer et al., 2024; Ma et al., 2024a; Chen et al., 2024). In contrast to caching methods, our proposed MoDiff introduces a novel and principled framework to leverage the computation redundancy between sampling steps through modulated computing.

**Post-Training Quantization.** Quantization aims to reduce inference costs by converting floating-point numbers into low-bit integers (Nagel et al., 2021; Yang et al., 2019) using scaling factors. Post-training quantization (PTQ) has emerged as a powerful approach due to its training-free nature, making it suitable for pre-trained models (Li et al., 2021). Several studies have explored the application of PTQ techniques to diffusion models (Li et al., 2023; Wang et al., 2024; Huang et al., 2024; Shang et al., 2023; He et al., 2023b; Zhao et al., 2025). For example, Q-Diffusion introduces a time-step-aware calibration data sampling mechanism tailored for diffusion models, achieving strong performance with 8-bit activations. However, a common issue is that existing methods struggle to quantize activations to

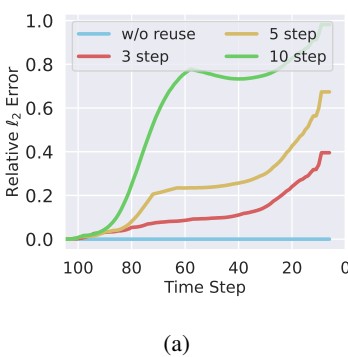
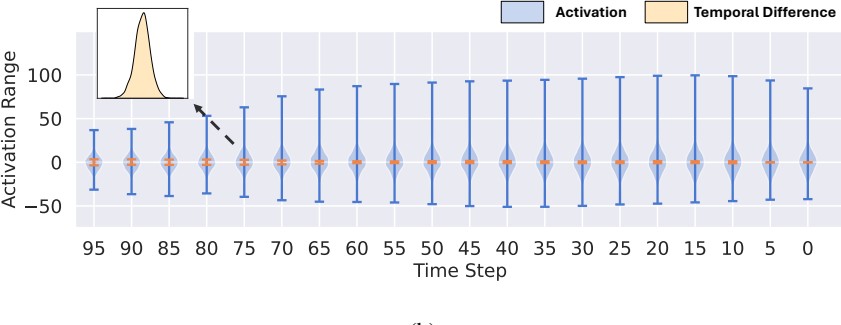

(a)    (b)

*Figure 1.* A preliminary study using DDIM on CIFAR-10 with 100 generation steps. (a) The relative $\ell_2$ distance between the cached and standard diffusion features in *middle block*, initialized from the same noise. As the reuse frequency increases, error accumulation becomes more significant. (b) The distribution of activations and their temporal differences across different diffusion time steps. The blue violin plots show that activation ranges fluctuate over time and exhibit outliers with long-tailed distributions. In contrast, the orange violin plots demonstrate more consistent ranges and concentrated distributions.

low bitwidths due to the presence of outlier values with large dynamic ranges (Xiao et al., 2023; Feng et al., 2024). One related method is PTQD (He et al., 2023b), which post processes quantized models to reduce the quantization error. We pose the detailed comparison to PTQD in Appendix E.

Another line of research focuses on quantization-aware training (QAT), which integrates the quantization process into the training phase, enabling model parameters to adapt to quantization (Nagel et al., 2022). These methods effectively address the challenges of low-bit quantization in diffusion models (Feng et al., 2024; He et al., 2023a). However, QAT approaches require costly retraining of diffusion models, which is **orthogonal to but not the focus of this work**. In contrast, the proposed MoDiff framework can be seamlessly integrated to state-of-the-art PTQ methods to reduce activation bit-widths without additional training.

## 3. Preliminary Study

In this section, we introduce the fundamental concepts of diffusion models and quantization. Additionally, we examine the challenges associated with existing caching and quantization methods with preliminary experiments.

### 3.1. Diffusion Models and Cache Reusing

Diffusion models consist of two processes: a forward process and a backward process, operating over $T$ steps. Using Denoising Diffusion Probabilistic Models (DDPMs) as an example (Ho et al., 2020), the forward process incrementally adds noise to the image at each step, gradually transforming the data distribution into a standard Gaussian distribution:

$$q(\mathbf{x}_t|\mathbf{x}_{t-1}) = N(\mathbf{x}_t; \sqrt{1-\beta_t}\mathbf{x}_{t-1}, \beta_t I). \quad (1)$$

Meanwhile, the reverse process progressively denoises the Gaussian distribution, reconstructing the original image dis-

tribution step by step with a denoising network:

$$p_\theta(\mathbf{x}_{t-1}|\mathbf{x}_t) = N(\mathbf{x}_{t-1}; \mu_\theta(\mathbf{x}_t, t), \sigma_t^2 I), \quad (2)$$

where $\mu_\theta(\mathbf{x}_t, t)$ is predicted by a neural network, while $\sigma_t$ is typically set to $\beta_t$. With this parametrization, the sampling process can be expressed as:

$$\mathbf{x}_{t-1} = \frac{1}{\sqrt{1-\beta_t}} \left( \mathbf{x}_t - \frac{\beta_t}{\sqrt{1-\bar{\alpha}_t}} \epsilon_\theta(\mathbf{x}_t, t) \right) + \sigma_t \mathbf{z}, \quad (3)$$

where $\bar{\alpha}_t = \prod_{i=1}^{t}(1-\beta_t)$, and $\epsilon_\theta(\mathbf{x}_t, t)$ represents a U-Net that predicts the noise. However, since generating samples requires predicting noise across $T$ steps, the diffusion process is computationally expensive for practical applications.

Existing approaches reuse historical computations to accelerate the sampling process by exploiting the similarities between features at adjacent time steps in diffusion models (Ma et al., 2024b; Wimbauer et al., 2024; Ma et al., 2024a). However, these caching methods directly reuse past information, which often cause approximation errors and deviate from the standard generation path of diffusion models. This discrepancy introduces errors at each reuse step, which accumulate over multiple iterations. As a result, these techniques require careful design of reuse schedules and even rely on retraining to tune models, necessitating expensive hyperparameter search.

To illustrate the impact of reuse schedules, we conduct a preliminary study, where we apply caching to the residual connections of the U-Net in DDIM on CIFAR-10 without tuning following Ma et al. (2024b). Specifically, we reuse the activations from the previous time step for $N-1$ steps and update them at every $N$-th step. We compare the relative $\ell_2$ distance between standard diffusion models and the variant with reused caching in *middle block*. As shown in Figure 1a, the relative error increases significantly as the number of reuse steps and the time steps grows.

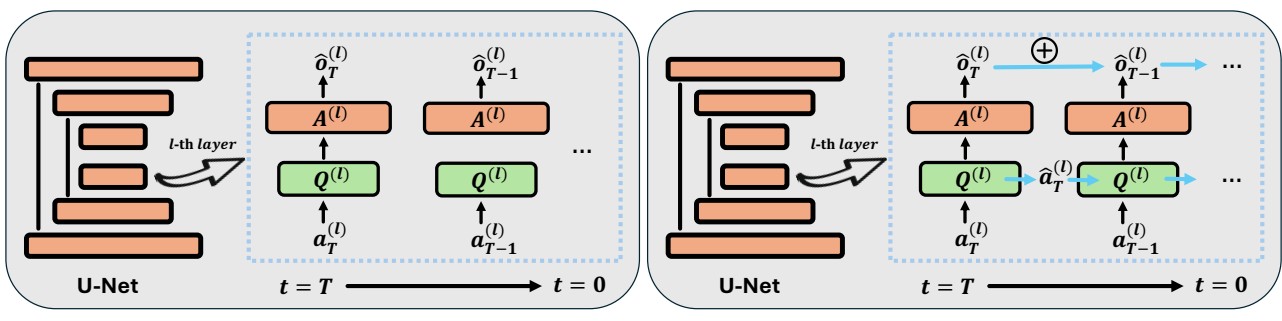

(a) Standard Quantization                    (b) Quantization w/ MoDiff

*Figure 2.* (a) Standard PTQ methods: The computations at different time steps are independent, with the raw activation $\mathbf{a}_t^{(l)}$ serving directly as the input to the quantizer. (b) Quantization with our MoDiff: For each linear operator, such as linear layers and convolutional layers, we cache the output from the previous time step, $\hat{\mathbf{a}}_t^{(l)}$, and input the temporal difference $\mathbf{a}_{t-1}^{(l)} - \hat{\mathbf{a}}_t^{(l)}$ into the quantizer. The final output is obtained by aggregating the current computation results of $\mathcal{A}^l$ with the cached output from the previous step $\hat{\mathbf{o}}_t^{(l)}$.

### 3.2. Post-Training Quantization

Quantization is an effective technique to reduce the inference cost of deep learning models by utilizing low-precision integers (Nagel et al., 2021). Given $\mathbf{x}$, we denote the integer representation as $\mathbf{x}_{\text{int}}$ and the dequantization vector as $Q(\mathbf{x})$:

$$\mathbf{x}_{\text{int}} = \text{clamp}(\left\lfloor \frac{\mathbf{x}}{s} \right\rceil + \mathbf{z}; 0, 2^b - 1), \qquad (4)$$

$$Q(\mathbf{x}) = s(\mathbf{x}_{\text{int}} - \mathbf{z}), \qquad (5)$$

where $b$ represents the quantization bandwidth, and $\text{clamp}(\cdot)$ enforces value cut-offs between two integer bounds. The parameters $s$ and $\mathbf{z}$ correspond to the scale factor and zero point, respectively. PTQ (Shang et al., 2023) dynamically estimates the parameters $s$ and $\mathbf{z}$ or derives them using calibration datasets with a pre-trained model. Due to its simplicity and efficiency, PTQ is widely adopted.

The major challenge for PTQ methods in diffusion models is to use low-bit quantization. First, the activation tensor ranges vary significantly across different time steps, as illustrated by the height of the blue violin plot in Figure 1b. This variation makes it difficult for a shared scaling parameter $s$ to handle all ranges. Second, significant outlier values exist within the activations at each time step. In Figure 1b, the width of the violin plot represents the distribution of activation values for a specific time step. These outliers make it challenging to select a scaling parameter $s$ that minimizes both clipping error and rounding error simultaneously.

In a nutshell, both caching and PTQ methods face their own inherent challenges, highlighting the need for a more effective strategy that can incorporate historical information while also mitigating the effects of activation distributions.

## 4. Modulated Diffusion

In this section, we propose Modulated Diffusion (MoDiff), a novel framework to accelerate all diffusion models with low-bit activation quantization, as shown in Figure 2. We introduce a high-level motivation in Section 4.1. To ease the understanding, we first present an equivalent reformulation of the diffusion process to reduce the quantization error in Section 4.2. Then, we propose novel error-compensated modulation to address accumulated error across the diffusion steps in Section 4.3. Computation and memory costs are discussed in Section 4.4, followed by theoretical error analyses of our MoDiff in Section 4.5.

### 4.1. High-Level Motivation

While the heuristic design of caching methods exhibit significant and accumulated computation errors, the motivation to leverage computation in previous time steps to reduce computation in future time steps is still of great interest. Inspired by the similarity of activation patterns across adjacent time steps, we measure the temporal differences between activation values over the diffusion process as $\mathbf{a}_t^{(l)} - \mathbf{a}_{t+1}^{(l)}$, where $\mathbf{a}_t^{(l)}$ represents the activation at time step $t$ for the $l$-th layer of the denoising network. The distribution of these differences is visualized in Figure 1b in orange color. Compared to the activations, their temporal differences exhibit a much smaller and more consistent range across time steps. Moreover, their distribution is more concentrated, effectively reducing the presence of outliers. These interesting observation and analyses suggest that a strong motivation to leverage this temporal stability with quantized computing to alleviate the computation errors in existing approaches.

### 4.2. Modulated Quantization

**Notation.** Let $\mathcal{A}^{(l)}(\cdot)$ denote the $l$-th linear operator in the denoising network, such as the linear and convolutional layers, where $l \in \{1, 2, \ldots, L\}$. We denote the input and output activations for $\mathcal{A}^{(l)}(\cdot)$ at time step $t$ as $\mathbf{a}_t^{(l)}$ and $\mathbf{o}_t^{(l)} = \mathcal{A}^{(l)}(\mathbf{a}_t^{(l)})$, respectively. Note that we focus on accel-

erating the computation of linear operators, such as linear and convolutional layers, since they are the most costly operations in neural networks and account for the majority of computation during data generation (Zhao et al., 2024).

Motivated by the insights from Section 4.1, we propose a novel modulated computation to reformulate the computation of each $l$-th linear layer in the denoising network in diffusion models as follows:

$$
\begin{cases}
\mathbf{o}_T^{(l)} & = \mathcal{A}^{(l)}(\mathbf{a}_T^{(l)}) \\
\mathbf{o}_{T-1}^{(l)} & = \mathcal{A}^{(l)}(\mathbf{a}_{T-1}^{(l)}) = \mathcal{A}^{(l)}(\mathbf{a}_{T-1}^{(l)} - \mathbf{a}_T^{(l)}) + \mathbf{o}_T^{(l)} \\
& \cdots \\
\mathbf{o}_t^{(l)} & = \mathcal{A}^{(l)}(\mathbf{a}_t^{(l)}) = \mathcal{A}^{(l)}(\mathbf{a}_t^{(l)} - \mathbf{a}_{t+1}^{(l)}) + \mathbf{o}_{t+1}^{(l)} \\
& \cdots \\
\mathbf{o}_1^{(l)} & = \mathcal{A}^{(l)}(\mathbf{a}_1^{(l)}) = \mathcal{A}^{(l)}(\mathbf{a}_1^{(l)} - \mathbf{a}_2^{(l)}) + \mathbf{o}_2^{(l)}
\end{cases}
\tag{6}
$$

where the output $\mathbf{o}_t^{(l)}$ in time step $t$ can be equivalently computed by incrementally refining the output $\mathbf{o}_{t+1}^{(l)}$ computed in previous time step $t+1$ with the modulated computation $\mathcal{A}^{(l)}(\mathbf{a}_t^{(l)} - \mathbf{a}_{t+1}^{(l)})$. Specifically, the second equality in each equation holds because of the linearity of the operator $\mathcal{A}^{(l)}$:

$$
\begin{aligned}
\mathcal{A}^{(l)}(\mathbf{a}_t^{(l)}) &= \mathcal{A}^{(l)}(\mathbf{a}_t^{(l)}) - \mathcal{A}^{(l)}(\mathbf{a}_{t+1}^{(l)}) + \mathcal{A}^{(l)}(\mathbf{a}_{t+1}^{(l)}) \\
&= \mathcal{A}^{(l)}(\mathbf{a}_t^{(l)} - \mathbf{a}_{t+1}^{(l)}) + \mathbf{o}_{t+1}^{(l)}.
\end{aligned}
$$

We further propose to apply a quantizer $Q$ to approximate the temporal difference before quantized computation[1]:

$$
\begin{cases}
\hat{\mathbf{o}}_T & = \mathcal{A}\big(Q(\mathbf{a}_T)\big) & \approx \mathcal{A}(\mathbf{a}_T) \\
\hat{\mathbf{o}}_{T-1} & = \mathcal{A}\big(Q(\mathbf{a}_{T-1} - \mathbf{a}_T)\big) + \hat{\mathbf{o}}_T & \approx \mathcal{A}(\mathbf{a}_{T-1}) \\
& \cdots & \\
\hat{\mathbf{o}}_t & = \mathcal{A}\big(Q(\mathbf{a}_t - \mathbf{a}_{t+1})\big) + \hat{\mathbf{o}}_{t+1} & \approx \mathcal{A}(\mathbf{a}_t) \\
& \cdots & \\
\hat{\mathbf{o}}_1 & = \mathcal{A}\big(Q(\mathbf{a}_1 - \mathbf{a}_2)\big) + \hat{\mathbf{o}}_2 & \approx \mathcal{A}(\mathbf{a}_1)
\end{cases}
\tag{7}
$$

Since the temporal difference $\mathbf{a}_t^{(l)} - \mathbf{a}_{t+1}^{(l)}$ has a much smaller and concentrated range as discussed in Section 4.1, its quantization will incur much smaller quantization errors.

*Remark* 4.1. When the input range falls bellow a tolerable threshold due to significant computation redundancy, MoDiff allows assigning a 0-bit representation in the quantizer, which skips the computation. This behavior subsumes existing heuristic caching strategies (Ma et al., 2024b) as special cases within a generalizable and principled framework, allowing more flexible control over caching.

---

[1]Note that the proposed modulated computation will be independently applied to every costly linear neural layer, so we omit the superscript index $(l)$ in the rest of the paper for simplicity.

## 4.3. Error-Compensated Modulation

While the modulated computation and quantization in Eq. (7) can reduce the activation quantization errors, comparing with the full-precision computation in Eq. (6), the computation error $\mathbf{o}_t - \hat{\mathbf{o}}_t$ will be carried over across the diffusion time steps and cause large accumulated errors. In this section, we introduce a novel error-compensated modulation to address the error accumulation, which leads to the complete **MoDiff framework** as follows:

$$
\begin{align}
\hat{\mathbf{a}}_T &= Q(\mathbf{a}_T) \tag{8} \\
\hat{\mathbf{o}}_T &= \mathcal{A}(\hat{\mathbf{a}}_T) \tag{9} \\
\hat{\mathbf{a}}_{T-1} &= Q(\mathbf{a}_{T-1} - \hat{\mathbf{a}}_T) + \hat{\mathbf{a}}_T \tag{10} \\
\hat{\mathbf{o}}_{T-1} &= \mathcal{A}(\hat{\mathbf{a}}_{T-1}) = \mathcal{A}\big(Q(\mathbf{a}_{T-1} - \hat{\mathbf{a}}_T)\big) + \hat{\mathbf{o}}_T \tag{11} \\
& \cdots \tag{12} \\
\hat{\mathbf{a}}_t &= Q(\mathbf{a}_t - \hat{\mathbf{a}}_{t+1}) + \hat{\mathbf{a}}_{t+1} \tag{13} \\
\hat{\mathbf{o}}_t &= \mathcal{A}(\hat{\mathbf{a}}_t) = \mathcal{A}\big(Q(\mathbf{a}_t - \hat{\mathbf{a}}_{t+1})\big) + \hat{\mathbf{o}}_{t+1} \tag{14} \\
& \cdots \tag{15} \\
\hat{\mathbf{a}}_1 &= Q(\mathbf{a}_1 - \hat{\mathbf{a}}_2) + \hat{\mathbf{a}}_2 \tag{16} \\
\hat{\mathbf{o}}_1 &= \mathcal{A}(\hat{\mathbf{a}}_1) = \mathcal{A}\big(Q(\mathbf{a}_1 - \hat{\mathbf{a}}_2)\big) + \hat{\mathbf{o}}_2 \tag{17}
\end{align}
$$

Specifically, we construct an intermediate variable $\hat{\mathbf{a}}_t$ to store the activation that is actually computed through quantization, which keeps track of the quantization errors:

$$
\begin{aligned}
\mathbf{e}_t &= (\mathbf{a}_t - \hat{\mathbf{a}}_{t+1}) - Q(\mathbf{a}_t - \hat{\mathbf{a}}_{t+1}) \\
&= (\mathbf{a}_t - \hat{\mathbf{a}}_{t+1}) - (\hat{\mathbf{a}}_t - \hat{\mathbf{a}}_{t+1}) = \mathbf{a}_t - \hat{\mathbf{a}}_t,
\end{aligned}
\tag{18}
$$

where the second equation comes from Eq. (13). Since we do not have access to the accurate $\mathbf{o}_t$ but only its approximation $\hat{\mathbf{o}}_t$, the incremental refinement will be on top of $\hat{\mathbf{o}}_t$. Given that $\hat{\mathbf{o}}_t = \mathcal{A}(\hat{\mathbf{a}}_t)$ is a feature tranformation of $\hat{\mathbf{a}}_t$, the residual should be computed based on $\hat{\mathbf{a}}_t$ instead of $\mathbf{a}_t$, which will compensate the errors and avoid error accumulation. Note that as shown in Eq. (14), $\hat{\mathbf{o}}_t = \mathcal{A}(\hat{\mathbf{a}}_t)$ only represents their relation, but the actual quantized computation is the following: $\hat{\mathbf{o}}_t = \mathcal{A}\big(Q(\mathbf{a}_t - \hat{\mathbf{a}}_{t+1})\big) + \hat{\mathbf{o}}_{t+1}$. A slight rewrite of this update clearly illustrates how quantization error is compensated across the diffusion time steps:

$$
\hat{\mathbf{o}}_t = \mathcal{A}\big(Q(\mathbf{a}_t - \mathbf{a}_{t+1} + \mathbf{e}_{t+1})\big) + \mathbf{o}_{t+1} - \mathcal{A}(\mathbf{e}_{t+1}),
$$

where the previous time step misses the computation of $\mathcal{A}(\mathbf{e}_{t+1})$ but will be compensated in the next time step by adding $\mathbf{e}_{t+1}$ into the input activations.

*Remark* 4.2. The proposed MoDiff framework is agnostic to quantization methods and can be applied to existing PTQ methods. Therefore, it is orthogonal to the contributions of prior works in this area. Moreover, we argue that MoDiff is not limited to quantization. It can be extended to other techniques, such as sparse techniques (Han et al., 2015), further demonstrating its generality and versatility.

## 4.4. Computation and Memory Costs

**Computation Cost.** We categorize the computing operations into three main types: matrix multiplication, matrix addition, and quantization/dequantization. For matrix multiplication, our method maintains integer-only operations, identical to standard quantization techniques. Additionally, by reducing quantization errors, our approach enables the use of a lower bandwidth for activations, potentially reducing the computation cost. For matrix addition, our approach introduces two additional operations in $\mathbf{a}_t - \hat{\mathbf{a}}_{t+1}$ and $\hat{\mathbf{o}}_{t+1}$. For Quantization and Dequantization, only dequantization on $Q(\mathbf{a}_t - \hat{\mathbf{a}}_{t+1})$ is an additional step introduced by our approach. Modern quantization techniques (Nagel et al., 2021) indicate that matrix multiplication is the dominant computational cost during inference. Since our method introduces only a minimal number of additional additions and quantization/dequantization operations, their overhead is negligible in comparison to matrix multiplication. Consequently, our approach do not increase or even decrease the computation cost compared to existing PTQ methods.

**Memory Consumption.** One limitation of our method is that it requires additional memory to store the intermediate variable $\mathbf{a}_t$ and outputs $\mathbf{o}_t$ for each layer. However, as demonstrated in Section 5.3, this memory overhead remains negligible compared to the model size when using small batch sizes and low-bit activation quantization. Furthermore, we can locally select the layers that use MoDiff, allowing for a trade-off between performance and memory efficiency.

## 4.5. Theoretical Error Analysis

The proposed MoDiff framework enables quantization with low quantization error while mitigating error accumulation through error compensation. In this section, we provide theoretical analyses to formally justify these advantages. The following theorem establishes the relationship between input magnitude and quantization error. For simplicity, we use dynamic quantizers, which determine the scaling parameter based on the input values to avoid clipping errors, and we consider vector inputs instead of assuming a specific distribution for the input data.

**Theorem 4.3** (Quantization Error). *Let $\mathbf{x} \in \mathbb{R}^d$ be a vector, and let the quantization bandwidth be $b \in \mathbb{N}$. Define the max-min dynamic quantizer as follows:*

$$s = \frac{\max(\mathbf{x}) - \min(\mathbf{x})}{2^b - 1}, \quad (19)$$

$$\mathbf{z} = \left\lfloor -\frac{\min(\mathbf{x})}{s} \right\rceil, \quad (20)$$

$$\mathbf{x}_{int} = clamp(\left\lfloor \frac{\mathbf{x}}{s} \right\rceil + \mathbf{z}, 0, 2^b - 1). \quad (21)$$

*The corresponding dequantization is given by:*

$$Q(\mathbf{x}) = s(\mathbf{x}_{int} - \mathbf{z}). \quad (22)$$

*The quantization error is bounded in terms of the quantization scaling factor $s$, which depends on the range of $\mathbf{x}$ and the bandwidth $b$. Specifically, we have:*

$$\|\mathbf{x} - Q(\mathbf{x})\|_2^2 \leq s^2 d = \frac{(\max(\mathbf{x}) - \min(\mathbf{x}))^2 d}{(2^b - 1)^2}. \quad (23)$$

The proof is provided in Appendix A.1. Theorem 4.3 establishes that quantization error is directly influenced by the input range and quantization bandwidth. Specifically, for a smaller input range, lower-bit quantization can achieve the same error bound. Our preliminary results show that the residuals exhibit a significantly reduced activation range, more than $10\times$ smaller, which suggests that activation bit precision can be lowered by at least 3 bits while maintaining comparable quantization error.

To illustrate how error-compensated modulation eliminates error accumulation, we assume that the inputs are independent for simplicity. The following theorem demonstrates that it reduces error accumulation at an exponential rate:

**Theorem 4.4.** *Let $\mathcal{A}(\cdot)$ be a linear operator and consider a sequence of inputs $\mathbf{a}_T, \mathbf{a}_{T-1}, \ldots, \mathbf{a}_1$, with corresponding outputs $\mathbf{o}_T, \mathbf{o}_{T-1}, \ldots, \mathbf{o}_1$. Given a quantization operator $Q$, we estimate the outputs using standard modulation:*

$$\tilde{\mathbf{o}}_t = \mathcal{A}(Q(\mathbf{a}_t - \mathbf{a}_{t+1})) + \tilde{\mathbf{o}}_{t+1}, \quad (24)$$

$$\tilde{\mathbf{o}}_T = \mathcal{A}(\mathbf{a}_T), \quad (25)$$

*where $t = T-1, \ldots, 2, 1$. Similarly, we estimate the outputs using error-compensated modulation:*

$$\hat{\mathbf{o}}_t = \mathcal{A}(Q(\mathbf{a}_t - \hat{\mathbf{a}}_{t+1})) + \hat{\mathbf{o}}_{t+1}, \quad (26)$$

$$\hat{\mathbf{a}}_t = Q(\mathbf{a}_t - \hat{\mathbf{a}}_{t+1}) + \hat{\mathbf{a}}_{t+1}, \quad (27)$$

$$\hat{\mathbf{o}}_T = \mathcal{A}(\mathbf{a}_T), \quad \hat{\mathbf{a}}_T = \mathbf{a}_T, \quad (28)$$

*where $t = T - 1, \ldots, 2, 1$. Suppose the quantization operator $Q$ satisfies the following error bound:*

$$\|\mathbf{x} - Q(\mathbf{x})\|_2^2 \leq c\|\mathbf{x}\|_2^2, \quad 0 < c < 1. \quad (29)$$

*Then, the estimation errors are bounded as follows:*

$$\|\mathbf{o}_t - \tilde{\mathbf{o}}_t\|_2^2 \leq \sum_{k=t}^{T-1} 2^{T-k-1} c\|\mathcal{A}\|_2^2 \|\mathbf{a}_k - \mathbf{a}_{k+1}\|_2^2, \quad (30)$$

$$\|\mathbf{o}_t - \hat{\mathbf{o}}_t\|_2^2 \leq \sum_{k=t}^{T-1} (2c)^{T-k-1} \|\mathcal{A}\|_2^2 \|\mathbf{a}_k - \mathbf{a}_{k+1}\|_2^2. \quad (31)$$

The proof is provided in Appendix A.2. Here, we assume that the quantization error is bounded by the input magnitude with a coefficient smaller than $1/2$, which is a direct corollary of Theorem 4.3 with appropriate $b$ as shown in Appendix A.3. Theorem 4.4 provides two key insights. First,

*Table 1.* The IS, FID, sFID, and GBOPs for CIFAR-10 with DDIM under different precisions. The best performance is **bolded**.

| Methods | Bits (W/A) | GBops | IS ↑ | FID ↓ | sFID ↓ | Bits (W/A) | GBops | IS ↑ | FID ↓ | sFID ↓ |
|---|---|---|---|---|---|---|---|---|---|---|
| Full Prec. (Act) | 8/32 | 1636 | 9.00 | 4.24 | 4.41 | 4/32 | 818 | 8.78 | 5.09 | 5.19 |
| Q-Diff | | | **9.48** | 3.75 | 4.49 | | | **9.12** | **4.93** | 5.03 |
| Q-Diff+MoDiff (Ours) | 8/8 | 409 | 9.10 | 4.10 | 4.39 | 4/8 | 204 | 9.08 | 5.13 | 5.18 |
| LCQ | | | 9.01 | 4.21 | 4.41 | | | 8.80 | 4.96 | **4.94** |
| LCQ+MoDiff (Ours) | | | 9.10 | 4.10 | **4.39** | | | 9.08 | 4.95 | 4.95 |
| Q-Diff | | | 8.76 | 29.16 | 13.81 | | | 8.51 | 28.60 | 15.09 |
| Q-Diff+MoDiff (Ours) | 8/6 | 307 | **9.38** | 4.19 | **4.32** | 4/6 | 153 | 8.85 | 5.62 | 4.93 |
| LCQ | | | 9.24 | **4.15** | 4.61 | | | **9.01** | **4.49** | 4.94 |
| LCQ+MoDiff (Ours) | | | 9.01 | 4.21 | 4.40 | | | 8.80 | 5.01 | **4.92** |
| Q-Diff | | | 2.19 | 332.75 | 100.37 | | | 2.47 | 325.76 | 92.84 |
| Q-Diff+MoDiff(Ours) | 8/4 | 205 | 9.71 | 13.41 | 11.25 | 4/4 | 102 | 9.60 | 13.62 | 11.94 |
| LCQ | | | **10.01** | 24.09 | 13.07 | | | **9.72** | 22.50 | 12.95 |
| LCQ+MoDiff (Ours) | | | 9.08 | **4.31** | **4.38** | | | 8.82 | **5.10** | **4.94** |
| Q-Diff | | | 1.19 | 457.35 | 165.79 | | | 1.19 | 457.35 | 165.79 |
| Q-Diff+MoDiff (Ours) | 8/3 | 153 | 5.19 | 90.34 | 41.26 | 4/3 | 77 | 7.34 | 47.35 | 13.87 |
| LCQ | | | 4.06 | 143.39 | 33.97 | | | 3.86 | 146.29 | 33.56 |
| LCQ+MoDiff (Ours) | | | **9.02** | **4.14** | **4.38** | | | **8.79** | **4.98** | **4.95** |
| Q-Diff | | | 1.19 | 457.34 | 165.79 | | | 1.19 | 457.34 | 165.79 |
| Q-Diff+MoDiff (Ours) | 8/2 | 102 | 1.82 | 266.68 | 75.88 | 4/2 | 51 | 1.36 | 387.75 | 168.38 |
| LCQ | | | 1.20 | 429.59 | 146.46 | | | 1.20 | 430.26 | 146.91 |
| LCQ+MoDiff (Ours) | | | **8.94** | **15.85** | **8.42** | | | **8.63** | **18.10** | **11.02** |

MoDiff without error compensation accumulates error more than linearly over the generation steps, making its performance highly dependent on the quantization parameter $c$. Second, error-compensation modulation in MoDiff ensures that errors from previous time steps are reduced exponentially, preventing error accumulation.

Finally, we note that Theorem 4.4 assumes independent inputs. However, in diffusion models, $\mathbf{a}_t$ is computed layer by layer using $\mathbf{o}_t$, which can further accumulate errors. As a result, error compensation has greater meaning in the application of diffusion models compared to standard modulation, as counteracting error propagation is more indispensable.

## 5. Experiments

In this section, we first introduce the experimental setup. We then evaluate our method across different quantization precisions, demonstrating its ability to significantly reduce activation bit requirements compared to existing methods across multiple datasets. Additionally, we conduct comprehensive ablation studies and present visualizations of generated images to assess the effectiveness of MoDiff.

### 5.1. Experiment Settings

**Datasets, Models, and Evaluation.** We majorly evaluate the effectiveness of our MoDiff on the CIFAR-10 ($32 \times 32$), LSUN-Bedrooms ($256 \times 256$), and LSUN-Church-Outdoor ($256 \times 256$) datasets (Krizhevsky et al., 2009; Yu et al., 2015). For CIFAR-10, we use DDIM models with 100 denoising steps (Song et al., 2021a). For the LSUN datasets, we use Latent Diffusion Models with downsampling factors

of 4 and 8, referred to as LDM-4 (Bedrooms) and LDM-8 (Churches), respectively (Rombach et al., 2022). We use 500 sampling steps for LDM-4 and 200 steps for LDM-8. To demonstrate the generalization capability of MoDiff across datasets and architecture, we also conduct experiments on Stable Diffusion and Transformer-based models (Peebles & Xie, 2023) on MS-COCO (Lin et al., 2014) and ImageNet (Russakovsky et al., 2015), respectively. Additional details and results are provided in Appendix C.1 and C.2.

We assess generation quality using Inception Score (IS) (Salimans et al., 2016), Fréchet Inception Distance (FID) (Heusel et al., 2017), and Sliced Fréchet Inception Distance (sFID) (Salimans et al., 2016) for CIFAR-10, and FID and sFID for the LSUN, as IS is not a reliable metric for datasets that significantly deviate from ImageNet categories. All metrics are computed based on 50,000 generated images. Additionally, we provide precision and recall measurements (Sajjadi et al., 2018) in Appendix C.3.

**Quantization Methods.** We use dynamic channel-wise quantization and Q-Diffusion as the base quantization methods and apply MoDiff to both (Dettmers et al., 2022; Li et al., 2023). We also present results using dynamic tensor-wise quantization in Appendix D.2. Additionally, we include results with full-precision activation (32 bits), for comparison. For weight quantization, we adopt the MSE reconstruction method, following the Q-Diffusion checkpoints. For activation quantization, dynamic channel-wise quantization determines the scaling factor based on the channel-wise min-max range of the input. Due to its dynamic nature, we directly apply MoDiff to this method. In contrast, Q-Diffusion optimizes the scaling factor by minimizing the

*Table 2.* The IS, FID, sFID, and GBOPs for LSUN-Church with LDM-8 under different precisions.

| Methods | Bits (W/A) | GBops | FID ↓ | sFID ↓ |
|---|---|---|---|---|
| Full Prec. (Act) | 8/32 | 5015 | 4.03 | 10.89 |
| Q-Diff | | | 4.24 | 10.57 |
| Q-Diff+MoDiff (Ours) | 8/8 | 1254 | **3.85** | 10.82 |
| LCQ | | | 4.02 | 11.53 |
| LCQ+MoDiff (Ours) | | | 3.99 | **10.06** |
| Q-Diff | | | 55.13 | 30.98 |
| Q-Diff+MoDiff (Ours) | 8/6 | 940 | 5.43 | 13.41 |
| LCQ | | | 4.50 | 12.90 |
| LCQ+MoDiff (Ours) | | | **3.89** | **10.12** |
| Q-Diff | | | 355.85 | 187.56 |
| Q-Diff+MoDiff (Ours) | 8/4 | 627 | **3.97** | 11.16 |
| LCQ | | | 198.37 | 161.03 |
| LCQ+MoDiff (Ours) | | | 34.02 | **10.59** |
| Q-Diff | | | 367.51 | 354.59 |
| Q-Diff+MoDiff (Ours) | 8/3 | 470 | **5.40** | **13.81** |
| LCQ | | | 341.62 | 407.68 |
| LCQ+MoDiff (Ours) | | | 12.05 | 35.29 |

*Table 3.* The IS, FID, sFID, and GBOPs for LSUN-Bedrooms with LDM-4 under different precisions.

| Methods | Bits (W/A) | GBops | FID ↓ | sFID ↓ |
|---|---|---|---|---|
| Full Prec. | 8/32 | 25560 | 3.45 | 8.45 |
| LCQ | 8/8 | 6390 | 3.61 | 8.65 |
| LCQ+MoDiff (Ours) | | | **3.57** | **8.44** |
| LCQ | 8/6 | 4609 | 64.17 | 63.18 |
| LCQ+MoDiff (Ours) | | | **3.57** | **6.53** |
| LCQ | 8/4 | 3195 | 372.30 | 262.11 |
| LCQ+MoDiff (Ours) | | | **27.88** | **77.85** |

MSE reconstruction loss using calibration datasets across different time steps. To apply MoDiff to Q-Diffusion, we calibrate the activation quantizers by inputting the calibration datasets into our MoDiff and learn the scaling factors with the residual. Additional implementation details can be found in Appendix B. We also perform few-step generation experiments using MixDQ (Zhao et al., 2024) as the baseline, with results provided in Appendix D.4.

For quantization hyperparameters, we select weight bit widths from $\{4, 8\}$ and activation bit widths from $\{2, 3, 4, 6, 8\}$. For notation simplicity, we use the format **"W/A"**, where "W" represents the weight precision and "A" represents the activation precision. We refer to Q-Diffusion and dynamic channel-wise quantization as **LCQ** and **Q-Diff**, respectively. Our implementations based on them are denoted as **LCQ+MoDiff** and **Q-Diff+MoDiff**. We denote full-precision activation models as **Full Prec. (Act)**.

*Remark* 5.1. The primary objective of this paper is to demonstrate the effectiveness of our method. We do not report the real acceleration metrics, such as running time. Following existing works (Li et al., 2023; Wang et al., 2024), we evaluate efficiency by measuring the number of binary operations (Bops) per denoising step for a single image with the help of DeepSpeed (Song et al., 2023). Implementing acceleration on specialized hardware is beyond the scope of this work, but will be a promising future direction, which is plausible given the increasing hardware support for low-precision formats such as 4-bit integers (Dave et al., 2019).

### 5.2. Main Results on CIFAR10 and LSUN

We conducted experiments to generate images using quantized diffusion models and measure their quality. The IS, FID, and sFID scores for CIFAR-10, Churches, and Bedrooms are presented in Tables 1, 2, and 3, respectively. Due

to page limitations, we only present results for 8-bit weight quantization on the LSUN datasets here. For results with 4-bit weight quantization, please refer to Appendix D.1. We highlight the best performance in bold. Based on these results, we draw the following conclusions.

**Generation Quality.** Our method preserves high generation quality and significantly outperforms the base quantization approach when using lower activation precision across different quantization methods and datasets. Specifically, with LCQ+MoDiff, activation precision in dynamic quantization can be reduced to 3 bits for CIFAR-10 without sacrificing generation quality. In contrast, the base quantization method experiences a significant performance drop even at 6-bit activation precision. For example, the sFID score of Q-Diff on CIFAR-10 degrades from 4.49 to 13.81. Furthermore, even at 2-bit activation precision, our method maintains an sFID of 8.42 on CIFAR-10.

**Generality.** Our method is generalizable across different datasets and various quantization methods. For both Q-Diff and LCQ, our approach consistently improves their performance. However, we observe that quantizing activations to extremely low-bit precision becomes increasingly challenging for higher-resolution datasets, even with our method. As shown in Tables 2 and 3, the FID and sFID scores increase significantly at 3-bit precision for the Churches dataset and become notably high at 4-bit precision for the Bedrooms dataset. This challenge arises because LDMs are deeper and contain higher-dimensional hidden embeddings, making it more difficult to minimize quantization error.

**Efficiency.** As shown in Tables 1, 2, and 3, our method consistently reduces binary operations (Bops) for generation. For a full-precision activation model, one denoising step in CIFAR-10 requires 1636 GBops. In contrast, LCQ+MoDiff completes inference with only 154 GBops without any performance degradation, achieving over $10\times$ computational savings. However, the generation quality (sFID) begins to degrade at 8/6-bit precision, which requires 307 GBops.

### 5.3. Ablation Study

In this section, we conduct ablation studies to analyze the effectiveness of MoDiff. First, we evaluate the impact of

*Table 4.* FID on CIFAR-10 using the DDIM sampler in the ablation study of error compensation. "EC" denotes error compensation and the best performance is **bolded**.

| Bits (W/A) | LCQ | LCQ+MoDiff w/o EC | w/ EC |
|---|---|---|---|
| 8/8 | 4.61 | 4.41 | **4.40** |
| 8/6 | 13.07 | 10.21 | **4.38** |
| 8/4 | 33.97 | 25.42 | **4.38** |

error-compensation modulation. Next, we demonstrate the compatibility of MoDiff with different samplers. Additionally, we examine how our method balances the trade-off between memory and computational efficiency. Finally, we present visualization results to further illustrate the effectiveness of MoDiff in Appendix F. To further demonstrate the generalization of MoDiff, we also include results on few-step diffusion models, as presented in the Appendix D.4.

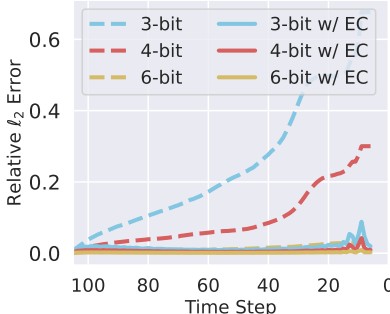

*Figure 3.* The relative $\ell_2$ distance between the features in the standard diffusion model and the quantized model in *middle block*. "w/ EC" denotes the use of the error-compensation technique.

**Effects of error compensation** We demonstrate how the error compensation technique mitigates error accumulation by comparing both generation quality and the quantization error, using DDIM on CIFAR-10 with LCQ. FIDs are shown in Table 4. To quantify quantization error, we compute the relative $\ell_2$ distance between the features of *middle block* in the standard diffusion model and the quantized model, both initialized from the same noise. As shown in Figure 3, the dashed line represents the relative error without error compensation, while the solid line represents the error with compensation. Without error compensation, error accumulation becomes significant below 6-bit precision. In contrast, with error compensation, the error remains minimal, even at 3-bit precision. In summary, our technique effectively avoids error accumulation in modulated computing.

**Different Samplers** We demonstrate that MoDiff is compatible with different samplers in diffusion models. While DDIM is used as the sampler in our main experiments, we also evaluate our method with DDPM on LCQ using the LSUN-Bedroom dataset, as shown in Table 5. Our results

*Table 5.* FID and sFID on LSUN-Bedrooms using the DDPM sampler under different quantization precisions with LCQ. The best performance is **bolded**.

| Methods | Bits (W/A) | FID ↓ | sFID ↓ |
|---|---|---|---|
| LCQ (DDPM) | 8/8 | 3.61 | 8.65 |
| LCQ+MoDiff (DDPM) | | **3.39** | **8.02** |
| LCQ (DDPM) | 8/6 | 50.17 | 52.18 |
| LCQ+MoDiff (DDPM) | | **12.60** | **13.71** |
| LCQ (DDPM) | 8/4 | 102.16 | 104.18 |
| LCQ+MoDiff (DDPM) | | **34.25** | **30.12** |

indicate that MoDiff enhances the generation quality of DDPM, particularly at lower activation bit widths. However, the improvement is less pronounced compared to DDIM. This is because the DDPM sampler introduces random noise at each step, increasing the difference between adjacent features. Consequently, this leads to larger residual magnitudes, which in turn amplify quantization errors. We provide additional experiments on PLMS (Liu et al., 2022) and DPM solver (Lu et al., 2022) in Appendix D.3.

**Memory Consumption.** As discussed in Section 4, our method reduces quantization error at the cost of slightly increased memory usage. In Table 6, we demonstrate that MoDiff significantly reduces Bops at lower bit precision while maintaining manageable memory overhead. The results show that the memory overhead is minimal—no more than 4 MB. For more details, we refer to Appendix D.6.

*Table 6.* The relationship between BOPs and memory usage of our method using DDIM on CIFAR-10 for generating a single image.

| Measurement | W8A2 | W8A4 | W8A8 | W8A32 |
|---|---|---|---|---|
| sFID | 8.42 | 4.38 | 4.39 | 4.41 |
| GBops | 102 | 205 | 409 | 1636 |
| Memory (Mb) | 35.28 | 36.4 | 38.89 | 35.09 |

## 6. Conclusion

In this paper, we propose MoDiff, a principled framework for accelerating generative modeling. Our preliminary studies reveal the challenges in caching and PTQ methods. To address these, we introduce modulated quantization and error compensation, which reduce quantization error and mitigate error accumulation. We provide theoretical analyses demonstrating the effectiveness of our approach. Experimental results show that MoDiff significantly enhances activation quantization, enabling PTQ methods to operate at bit-widths as low as 3 bits without performance degradation. One limitation is that MoDiff reduces computation at the cost of increased memory usage. Additionally, we evaluate acceleration based on theoretical computational complexity rather than real-world hardware speedup. We leave hardware implementation and further memory optimizations of our MoDiff for future work.

## Acknowledgments

Weizhi Gao, Zhichao Hou, and Dr. Xiaorui Liu are supported by the National Science Foundation (NSF) Award under grant number IIS-2443182, NSF National AI Research Resource Pilot Award, NCSU Data Science Academy Seed Grant Award, and NCSU Faculty Research and Professional Development Award.

## Impact Statement

**Ethical Considerations.** This work introduces modulated diffusion models to accelerate the generation process of diffusion models. We do not foresee any significant ethical concerns associated with this approach.

**Societal Impact.** Enhancing the efficiency of diffusion models can facilitate the broader adoption of generative AI, benefiting both AI research and hardware development. This advancement has the potential to contribute to more efficient and accessible AI-driven solutions.

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

## A. Proof

In this section, we provide proofs for the theorems presented in the main paper.

### A.1. Proof of Theorem 4.3

**Theorem A.1** (Restated, 4.3). *Let $\mathbf{x} \in \mathbb{R}^d$ be a vector, and let the quantization bandwidth be $b \in \mathbb{N}$. Define the max-min dynamic quantizer as follows:*

$$s = \frac{\max(\mathbf{x}) - \min(\mathbf{x})}{2^b - 1}, \tag{32}$$

$$\mathbf{z} = \left\lfloor -\frac{\min(\mathbf{x})}{s} \right\rceil, \tag{33}$$

$$\mathbf{x}_{int} = clamp(\left\lfloor \frac{\mathbf{x}}{s} \right\rceil + \mathbf{z}, 0, 2^b - 1). \tag{34}$$

*The corresponding dequantization is given by:*

$$Q(\mathbf{x}) = s(\mathbf{x}_{int} - \mathbf{z}). \tag{35}$$

*The quantization error is bounded in terms of the quantization scaling factor $s$, which depends on the range of $\mathbf{x}$ and the bandwidth $b$. Specifically, we have:*

$$\|\mathbf{x} - Q(\mathbf{x})\|_2^2 \le s^2 d = \frac{(\max(\mathbf{x}) - \min(\mathbf{x}))^2 d}{(2^b - 1)^2}. \tag{36}$$

*Proof:* For any $i \in \{1, 2, \ldots, d\}$, the value $x_i$ satisfies the following property:

$$0 \le \lfloor \frac{\min(\mathbf{x})}{s} \rfloor + \lfloor \frac{-\min(\mathbf{x})}{s} \rfloor \le \lfloor \frac{x_i}{s} \rfloor + z \le \lfloor \frac{\max(\mathbf{x})}{s} \rfloor + \lfloor \frac{-\min(\mathbf{x})}{s} \rfloor \le \frac{\max(\mathbf{x}) - \min(\mathbf{x})}{s}. \tag{37}$$

From condition ( 32), we have:

$$0 \le x_{\text{int}} \le 2^b - 1.$$

This ensures that the clipping error is zero, meaning we only need to consider the rounding error. Thus, we obtain:

$$x_i - Q(x)_i = x_i - \lfloor \frac{x_i}{s} \rceil s \tag{38}$$

$$= s \left( \frac{x_i}{s} - \lfloor \frac{x_i}{s} \rceil \right) \le s. \tag{39}$$

Therefore, applying this to the $\ell_2$-norm error bound, we derive:

$$\|\mathbf{x} - Q(\mathbf{x})\|_2^2 = \sum_{i=1}^{d} (x_i - Q(x)_i)^2 \tag{40}$$

$$\le \sum_{i=1}^{d} s^2 \tag{41}$$

$$= s^2 d. \tag{42}$$

$$\square$$

## A.2. Proof of Theorem 4.4

**Theorem A.2** (Restated, 4.4). *Let $\mathcal{A}(\cdot)$ be a linear operator and consider a sequence of inputs $\mathbf{a}_T, \mathbf{a}_{T-1}, \ldots, \mathbf{a}_1$, with corresponding outputs $\mathbf{o}_T, \mathbf{o}_{T-1}, \ldots, \mathbf{o}_1$. Given a quantization operator $Q$, we estimate the outputs using standard modulation:*

$$\tilde{\mathbf{o}}_t = \mathcal{A}(Q(\mathbf{a}_t - \mathbf{a}_{t+1})) + \tilde{\mathbf{o}}_{t+1}, \tag{43}$$

$$\tilde{\mathbf{o}}_T = \mathcal{A}(\mathbf{a}_T), \tag{44}$$

*where $t = T-1, \ldots, 2, 1$. Similarly, we estimate the outputs using error-compensated modulation:*

$$\hat{\mathbf{o}}_t = \mathcal{A}(Q(\mathbf{a}_t - \hat{\mathbf{a}}_{t+1})) + \hat{\mathbf{o}}_{t+1}, \tag{45}$$

$$\hat{\mathbf{a}}_t = Q(\mathbf{a}_t - \hat{\mathbf{a}}_{t+1}) + \hat{\mathbf{a}}_{t+1}, \tag{46}$$

$$\hat{\mathbf{o}}_T = \mathcal{A}(\mathbf{a}_T), \quad \hat{\mathbf{a}}_T = \mathbf{a}_T, \tag{47}$$

*where $t = T-1, \ldots, 2, 1$. Suppose the quantization operator $Q$ satisfies the following error bound:*

$$\|\mathbf{x} - Q(\mathbf{x})\|_2^2 \le c\|\mathbf{x}\|_2^2, \quad 0 < c < \frac{1}{2}. \tag{48}$$

*Then, the estimation errors are bounded as follows:*

$$\|\mathbf{o}_t - \tilde{\mathbf{o}}_t\|_2^2 \le \sum_{k=t}^{T-1} 2^{T-k-1} c\|\mathcal{A}\|_2^2 \|\mathbf{a}_k - \mathbf{a}_{k+1}\|_2^2, \tag{49}$$

$$\|\mathbf{o}_t - \hat{\mathbf{o}}_t\|_2^2 \le \sum_{k=t}^{T-1} (2c)^{T-k-1} \|\mathcal{A}\|_2^2 \|\mathbf{a}_k - \mathbf{a}_{k+1}\|_2^2. \tag{50}$$

*Proof:* Denote the error for standard modulation in Equation (43) as $\tilde{\mathbf{e}}_t$ and for error-compensation modulation in Equation (44) as $\hat{\mathbf{e}}_t$ at time step $t$. We first compute the error for standard modulation:

$$\tilde{\mathbf{e}}_t^2 = \|\mathbf{o}_t - \tilde{\mathbf{o}}_t\|_2^2 \tag{51}$$

$$= \|\mathbf{o}_t - \mathcal{A}(Q(\mathbf{a}_t - \mathbf{a}_{t+1})) - \tilde{\mathbf{o}}_{t+1}\|_2^2 \tag{52}$$

$$= \|\mathbf{o}_t - \mathbf{o}_{t+1} - \mathcal{A}(Q(\mathbf{a}_t - \mathbf{a}_{t+1})) + (\mathbf{o}_{t+1} - \tilde{\mathbf{o}}_{t+1})\|_2^2 \tag{53}$$

$$= \|\mathcal{A}(\mathbf{a}_t - \mathbf{a}_{t+1}) - \mathcal{A}(Q(\mathbf{a}_t - \mathbf{a}_{t+1})) + (\mathbf{o}_{t+1} - \tilde{\mathbf{o}}_{t+1})\|_2^2 \tag{54}$$

$$= \|\mathcal{A}(\mathbf{a}_t - \mathbf{a}_{t+1} - Q(\mathbf{a}_t - \mathbf{a}_{t+1})) + (\mathbf{o}_{t+1} - \tilde{\mathbf{o}}_{t+1})\|_2^2 \tag{55}$$

$$\le 2|\mathcal{A}(\mathbf{a}_t - \mathbf{a}_{t+1} - Q(\mathbf{a}_t - \mathbf{a}_{t+1}))\|_2^2 + 2\|\mathbf{o}_{t+1} - \tilde{\mathbf{o}}_{t+1}\|_2^2 \tag{56}$$

Since $\|(\mathbf{o}_{t+1} - \tilde{\mathbf{o}}_{t+1})\|_2^2$ represents the error from the previous time step, applying the submultiplicative inequality yields:

$$\tilde{\mathbf{e}}_t^2 = \|\mathbf{o}_t - \tilde{\mathbf{o}}_t\|_2^2 \tag{57}$$

$$\le 2\|\mathcal{A}\|_2^2 \|\mathbf{a}_t - \mathbf{a}_{t+1} - Q(\mathbf{a}_t - \mathbf{a}_{t+1})\|_2^2 + 2\mathbf{e}_{t+1}^2 \tag{58}$$

$$\le 2c\|\mathcal{A}\|_2^2 \|\mathbf{a}_t - \mathbf{a}_{t+1}\|_2^2 + 2\mathbf{e}_{t+1}^2, \tag{59}$$

Accumulating the error from time $T$ to $t$, we obtain Equation (49).

For the error-compensation modulation, we compute:

$$\hat{\mathbf{e}}_t^2 = \|\mathbf{o}_t - \hat{\mathbf{o}}_t\|_2^2 \tag{60}$$

$$= \|\mathbf{o}_t - \mathcal{A}(Q(\mathbf{a}_t - \hat{\mathbf{a}}_{t+1})) - \hat{\mathbf{o}}_{t+1}\|_2^2 \tag{61}$$

$$= \|\mathcal{A}(\mathbf{a}_t) - \mathcal{A}(Q(\mathbf{a}_t - \hat{\mathbf{a}}_{t+1})) - \mathcal{A}(\hat{\mathbf{a}}_{t+1})\|_2^2 \tag{62}$$

$$= \|\mathcal{A}(\mathbf{a}_t - \hat{\mathbf{a}}_{t+1} - Q(\mathbf{a}_t - \hat{\mathbf{a}}_{t+1}))\|_2^2 \tag{63}$$

$$\le c\|\mathcal{A}\|_2^2 \|\mathbf{a}_t - \hat{\mathbf{a}}_{t+1}\|_2^2 \tag{64}$$

Next, we expand $\mathbf{a}_t - \hat{\mathbf{a}}_{t+1}$:

$$\|\mathbf{a}_t - \hat{\mathbf{a}}_{t+1}\|_2^2 = \|\mathbf{a}_t - Q(\mathbf{a}_{t+1} - \hat{\mathbf{a}}_{t+2}) - \hat{\mathbf{a}}_{t+2}\|_2^2 \tag{65}$$

$$= \|\mathbf{a}_t - \mathbf{a}_{t+1} - Q(\mathbf{a}_{t+1} - \hat{\mathbf{a}}_{t+2}) + \mathbf{a}_{t+1} - \hat{\mathbf{a}}_{t+2}\|_2^2 \tag{66}$$

$$\leq 2\|\mathbf{a}_t - \mathbf{a}_{t+1}\|_2^2 + 2\|Q(\mathbf{a}_{t+1} - \hat{\mathbf{a}}_{t+2}) + \mathbf{a}_{t+1} - \hat{\mathbf{a}}_{t+2}\|_2^2 \tag{67}$$

$$\leq 2\|\mathbf{a}_t - \mathbf{a}_{t+1}\|_2^2 + 2c\|\mathbf{a}_{t+1} - \hat{\mathbf{a}}_{t+2}\|_2^2 \tag{68}$$

Substituting this into Equation (64), we complete the proof. $\square$

### A.3. Proof of Corollary

**Corollary A.3.** *Let $\mathbf{x} \in \mathbb{R}^d$ be a vector, and let the quantization bandwidth be $b \in \mathbb{N}$. Define the max-min dynamic quantizer as follows:*

$$s = \frac{\max(\mathbf{x}) - \min(\mathbf{x})}{2^b - 1}, \tag{69}$$

$$\mathbf{z} = \left\lfloor -\frac{\min(\mathbf{x})}{s} \right\rceil, \tag{70}$$

$$\mathbf{x}_{int} = clamp(\left\lfloor \frac{\mathbf{x}}{s} \right\rceil + \mathbf{z}, 0, 2^b - 1). \tag{71}$$

*The corresponding dequantization is given by:*

$$Q(\mathbf{x}) = s(\mathbf{x}_{int} - \mathbf{z}). \tag{72}$$

*For any $0 < c < \frac{1}{2}$, we can revise $Q$ with a new bandwidth $\hat{b}$ satisfying:*

$$\|\mathbf{x} - Q(\mathbf{x})\|_2^2 \leq c\|\mathbf{x}\|_2^2. \tag{73}$$

*Proof:* From Theorem 4.3, we have:

$$\|\mathbf{x} - Q(\mathbf{x})\|_2^2 \leq \frac{(\max(\mathbf{x}) - \min(\mathbf{x}))^2 d}{(2^b - 1)^2} \tag{74}$$

$$\leq \frac{4\|\mathbf{x}\|_\infty^2 d}{(2^b - 1)^2} \tag{75}$$

$$\leq \frac{4\|\mathbf{x}\|_2^2 d}{(2^b - 1)^2} \tag{76}$$

To satisfy the desired bound, we choose $\hat{b}$ such that:

$$\hat{b} \geq \log_2\left(\sqrt{\frac{4d}{c}} + 1\right). \tag{77}$$

Thus, the proof is complete. $\square$

## B. Implementation details

In this section, we talk about the hyperparameters in our experiments and the implementation details of our MoDiff.

**Baselines.** For the implementation of baselines, we follow the existing codebase. Specifically, we conduct Q-Diffusion experiments by directly using their provided code (Li et al., 2023). We also utilize the calibration datasets they provide to quantize the models at different bit levels. For LCQ, we follow the BRECQ framework and adopt channel-wise quantization (Li et al., 2021).

**MoDiff.** For our MoDiff implementation, we incorporate several key techniques:

- Bias Removal: We remove all bias terms from layers that apply MoDiff. This is necessary because our method, as described in Equation (13), requires layers to be bias-free to prevent unwanted accumulation of bias terms.

- Warm-up: We apply warm-up at the first step, where we use full activation for computation. More detailed analysis is shown in Appendix D.5.

- Calibration Dataset Reconstruction: We reconstruct the calibration dataset for Q-Diff + MoDiff, ensuring it captures nearby information. During calibration, we store the inputs and outputs of MoDiff rather than the raw activations.

- Layer-wise Reconstruction: Instead of reconstructing entire blocks, we reconstruct each layer individually, as we find this approach leads to more stable performance.

- Hyperparameter Consistency: We do not fine-tune the calibration hyperparameters, as optimizing them is not the primary focus of our work.

## C. Additional Main Results

### C.1. Results on Stable Diffusion

To demonstrate that our method generalizes to larger-scale datasets and higher resolutions, we conduct experiments on MS-COCO 2014 (Lin et al., 2014) using Stable Diffusion v1.4 with DPM solvers(Lu et al., 2022). We apply tensor-wise dynamic quantization and evaluate the quantized models within the Q-Diffusion framework. A total of 30,000 images are generated using 50 sampling steps. As shown in Table 7, the resulting FID scores confirm that MoDiff consistently performs well on large-scale diffusion models.

Table 7. The FID and sFID on MS-COCO with Stable Diffusion using PLMS solver under different precisions. The best performance is **bolded**.

| Methods | Bits (W/A) | FID ↓ | sFID ↓ |
|---|---|---|---|
| LTQ | 8/8 | 12.15 | 19.05 |
| LTQ+MoDiff (Ours) | | **12.14** | **19.05** |
| LTQ | 8/6 | 71.38 | 59.74 |
| LTQ+MoDiff (Ours) | | **13.21** | **20.07** |
| LTQ | 8/4 | 408.42 | 199.59 |
| LTQ+MoDiff (Ours) | | **225.22** | **104.12** |

### C.2. Results on Transformer-Based Models

To evaluate the generalizability of MoDiff across different architectures, we conduct experiments on the Diffusion Transformer (Peebles & Xie, 2023). Following PTQ4DiT (Wu et al., 2024), we use DiT-XL/2 as the baseline model. The experiments are performed on the ImageNet 256×256 dataset (Russakovsky et al., 2015) using tensor-wise dynamic quantization. We generate 10,000 images using 50 sampling steps for evaluation. As shown in Table 8, MoDiff consistently enhances generation quality under low activation bit widths.

Table 8. The IS, FID, and sFID for ImageNet 256x256 with DiT-XL/2 under different precisions. The best performance is **bolded**.

| Methods | Bits (W/A) | IS ↑ | FID ↓ | sFID ↓ |
|---|---|---|---|---|
| PTQ4DiT | 8/8 | 36.91 | 54.80 | 89.60 |
| PTQ4DiT+MoDiff (Ours) | | **37.37** | **53.76** | **89.53** |
| PTQ4DiT | 8/6 | 3.41 | 200.26 | 373.71 |
| PTQ4DiT+MoDiff (Ours) | | **36.74** | **54.74** | **88.49** |
| PTQ4DiT | 8/4 | 1.45 | 271.87 | 207.59 |
| PTQ4DiT+MoDiff (Ours) | | **17.23** | **90.91** | **102.07** |

## C.3. More Measurements on Generation Quality

In the main paper, we evaluate the quality of generated outputs using Inception Score (IS), Fréchet Inception Distance (FID), and sFID. Here, we further assess the performance of our method using precision and recall.

The results are presented in Table 9, Table 10, and Table 11. These results demonstrate that MoDiff effectively preserves precision and recall even at low activation bit levels. For instance, on CIFAR-10, LCQ+MoDiff achieves a precision of 0.58 and a recall of 0.50, whereas LCQ alone results in 0 for both metrics.

*Table 9.* The Precision and Recall for CIFAR-10 with DDIM under different Bits. The best performance is **bolded**.

| Methods | Bits (W/A) | Precision | Recall | Bits (W/A) | Precision | Recall |
|---|---|---|---|---|---|---|
| Full Prec. (Act) | 8/32 | 0.65 | 0.55 | 4/32 | 0.64 | 0.56 |
| Q-Diff | 8/8 | 0.65 | 0.55 | 4/8 | 0.66 | 0.58 |
| Q-Diff+MoDiff (Ours) | | 0.65 | 0.56 | | 0.65 | 0.58 |
| LCQ | | 0.67 | 0.59 | | 0.67 | 0.57 |
| LCQ+MoDiff (Ours) | | 0.66 | 0.59 | | 0.67 | 0.55 |
| Q-Diff | 8/6 | 0.46 | 0.47 | 4/6 | 0.47 | 0.44 |
| Q-Diff+MoDiff (Ours) | | 0.66 | 0.57 | | 0.65 | 0.59 |
| LCQ | | 0.67 | 0.58 | | 0.67 | 0.57 |
| LCQ+MoDiff (Ours) | | 0.66 | 0.58 | | 0.67 | 0.56 |
| Q-Diff | 8/4 | 0.08 | 0.00 | 4/4 | 0.05 | 0.00 |
| Q-Diff+MoDifff(Ours) | | 0.54 | 0.53 | | 0.53 | 0.55 |
| LCQ | | 0.47 | 0.44 | | 0.48 | 0.43 |
| LCQ+MoDiff (Ours) | | 0.67 | 0.59 | | 0.67 | 0.57 |
| Q-Diff | 8/3 | 0.00 | 0.00 | 4/3 | 0.00 | 0.00 |
| Q-Diff+MoDiff (Ours) | | 0.45 | 0.39 | | 0.33 | 0.32 |
| LCQ | | 0.33 | 0.08 | | 0.35 | 0.08 |
| LCQ+MoDiff (Ours) | | 0.66 | 0.59 | | 0.67 | 0.57 |
| Q-Diff | 8/2 | 0.00 | 0.00 | 4/2 | 0.00 | 0.00 |
| Q-Diff+MoDiff (Ours) | | 0.00 | 0.00 | | 0.14 | 0.00 |
| LCQ | | 0.00 | 0.00 | | 0.00 | 0.00 |
| LCQ+MoDiff (Ours) | | 0.58 | 0.50 | | 0.58 | 0.47 |

*Table 10.* The Precision and Recall for Church with LDM-8 under different Bits. The best performance is **bolded**.

| Methods | Bits (W/A) | Precision | Recall | Bits (W/A) | Precision | Recall |
|---|---|---|---|---|---|---|
| Full Prec. (Act) | 8/32 | 0.63 | 0.51 | 4/32 | 0.63 | 0.52 |
| LCQ | 8/8 | 0.62 | 0.47 | 4/8 | 0.62 | 0.46 |
| LCQ+MoDiff (Ours) | | 0.63 | 0.53 | | 0.63 | 0.53 |
| LCQ | 8/6 | 0.59 | 0.46 | 4/6 | 0.59 | 0.45 |
| LCQ+MoDiff (Ours) | | 0.63 | 0.53 | | 0.63 | 0.53 |
| LCQ | 8/4 | 0.03 | 0.14 | 4/4 | 0.02 | 0.07 |
| LCQ+MoDiff (Ours) | | 0.63 | 0.53 | | 0.63 | 0.5 |
| LCQ | 8/3 | 0.00 | 0.00 | 4/3 | 0.00 | 0.00 |
| LCQ+MoDiff (Ours) | | 0.61 | 0.34 | | 0.60 | 0.34 |

# D. Ablation Study

## D.1. Results on Other Weight Precision

In the main paper, we present results for 8-bit weight quantization on LSUN-Churches and LSUN-Bedroom for the page limitation. In this section, we extend our analysis to 4-bit weight quantization and observe consistent conclusions. As shown in Table 12 and Table 13, our method successfully maintains generation quality at 4/3 bits for Churches and 4/4 bits for Bedrooms. In contrast, LCQ experiences a significant performance drop.

*Table 11.* The Precision and Recall for Bedroom with LDM-4 under different Bits. The best performance is **bolded**.

| Methods | Bits (W/A) | Precision | Recall | Bits (W/A) | Precision | Recall |
|---|---|---|---|---|---|---|
| Full Prec. (Act) | 8/32 | 0.65 | 0.45 | 4/32 | 0.66 | 0.41 |
| LCQ | 8/8 | 0.65 | 0.45 | 4/8 | 0.68 | 0.41 |
| LCQ+MoDiff (Ours) | | 0.60 | 0.51 | | 0.62 | 0.47 |
| LCQ | 8/6 | 0.17 | 0.13 | 4/6 | 0.63 | 0.43 |
| LCQ+MoDiff (Ours) | | 0.59 | 0.51 | | 0.62 | 0.47 |
| LCQ | 8/4 | 0.00 | 0.00 | 4/4 | 0.00 | 0.00 |
| LCQ+MoDiff (Ours) | | 0.40 | 0.17 | | 0.46 | 0.22 |

*Table 12.* The IS, FID, sFID, and GBOPs for LSUN-Church with LDM under 4-bit weight quantization. The best performance is **bolded**.

| Methods | Bits (W/A) | GBops | FID ↓ | sFID ↓ |
|---|---|---|---|---|
| Full Prec. (Act) | 8/32 | 5015 | 4.03 | 10.89 |
| LCQ | 8/8 | 1254 | 4.02 | 11.53 |
| LCQ+MoDiff (Ours) | | | 3.99 | 10.06 |
| LCQ | 8/6 | 940 | 4.50 | 12.90 |
| LCQ+MoDiff (Ours) | | | 3.89 | 10.12 |
| LCQ | 8/4 | 627 | 198.37 | 161.03 |
| LCQ+MoDiff (Ours) | | | 34.02 | 10.59 |
| LCQ | 8/3 | 470 | 341.62 | 407.68 |
| LCQ+MoDiff (Ours) | | | 12.05 | 35.29 |

*Table 13.* The IS, FID, sFID, and GBOPs for LSUN-Bedrooms with LDM under 4-bit weight quantization. The best performance is **bolded**.

| Methods | Bits (W/A) | GBops | FID ↓ | sFID ↓ |
|---|---|---|---|---|
| Full Prec. | 8/32 | 25560 | 3.45 | 8.45 |
| LCQ | 8/8 | 6390 | 3.61 | 8.65 |
| LCQ+MoDiff (Ours) | | | 3.57 | 8.44 |
| LCQ | 8/6 | 4609 | 64.17 | 63.18 |
| LCQ+MoDiff (Ours) | | | 3.57 | 6.53 |
| LCQ | 8/4 | 3195 | 372.30 | 262.11 |
| LCQ+MoDiff (Ours) | | | 27.88 | 77.85 |

## D.2. Results on Tensor-Wise Quantization

In our main experiments, we present results using dynamic channel-wise quantization (LCQ). In this section, we extend our analysis to dynamic tensor-wise quantization (LTQ), which is more hardware-friendly. We conduct experiments on CIFAR-10 using DDIM, while continuing to use Q-Diffusion checkpoints for weight quantization. As shown in Table 14, our MoDiff framework is also effective for LTQ. However, the minimum activation bit-width achievable with LTQ is higher than that of LCQ. This is because tensor-wise quantization operates on higher-dimensional data, making accurate quantization more challenging.

## D.3. Results on More Samplers

In the main paper, we demonstrate that MoDiff generalizes to the DDPM sampler. Here, we further show its applicability to additional solvers. Specifically, we perform tensor-wise dynamic quantization using DPM-Solver-2 (Lu et al., 2022) on CIFAR-10 with 20 sampling steps. Additionally, we evaluate MoDiff with the PLMS solver using 50 steps on Stable Diffusion with the MS-COCO 2014 dataset (Liu et al., 2022). As shown in Table 15 and Table 7, MoDiff consistently

*Table 14.* The IS, FID, sFID, and GBOPs for CIFAR-10 with DDIM using tensor-wise quantization under different precisions. The best performance is **bolded**.

| Methods | Bits (W/A) | IS ↑ | FID ↓ | sFID ↓ | Bits (W/A) | IS ↑ | FID ↓ | sFID ↓ |
|---|---|---|---|---|---|---|---|---|
| Full Prec. (Act) | 8/32 | 9.00 | 4.24 | 4.41 | 4/32 | 8.78 | 5.09 | 5.19 |
| LTQ | 8/8 | **9.08** | **4.19** | 4.40 | 4/8 | **8.80** | **5.02** | 5.21 |
| LTQ+MoDiff (Ours) | | 9.04 | 4.21 | **4.37** | | 8.76 | 5.05 | **5.16** |
| LTQ | 8/6 | 8.98 | 9.93 | 8.69 | 4/6 | 8.89 | 9.96 | 8.07 |
| LTQ+MoDiff (Ours) | | **9.09** | **4.00** | **4.27** | | **8.80** | **5.04** | **4.42** |
| LTQ | 8/4 | 2.27 | 306.06 | 94.28 | 4/4 | 2.37 | 294.88 | 90.91 |
| LTQ+MoDiff(Ours) | | **8.37** | **28.19** | **19.90** | | **8.35** | **26.17** | **18.94** |
| LTQ | 8/2 | 1.19 | 457.25 | 165.85 | 4/2 | 1.19 | 457.11 | 165.61 |
| LTQ+MoDiff (Ours) | | **4.26** | **186.04** | **86.73** | | **3.29** | **146.52** | **87.78** |

improves FID scores across different solvers.

*Table 15.* The FID on CIFAR-10 with DDIM using DPM solver under different precisions. The best performance is **bolded**.

| Methods | Bits (W/A) | FID ↓ |
|---|---|---|
| DPM | 8/8 | 3.92 |
| DPM+MoDiff (Ours) | | **3.91** |
| DPM | 8/6 | 10.82 |
| DPM+MoDiff (Ours) | | **3.91** |
| DPM | 8/4 | 299.72 |
| DPM+MoDiff (Ours) | | **26.54** |

### D.4. Results on Fewer Generation Steps

To demonstrate that MoDiff remains effective with fewer generation steps, we conduct experiments on CIFAR-10 using the DDIM sampler with only 20 steps. Tensor-wise dynamic quantization is applied throughout. As shown in Table 16, MoDiff maintains strong performance even under this reduced-step setting.

*Table 16.* FID on CIFAR-10 using the DDIM sampler in the ablation study of fewer steps.

| Methods | Bits (W/A) | FID ↓ |
|---|---|---|
| LTQ | 8/8 | 6.93 |
| LTQ+MoDiff (Ours) | | **6.90** |
| LTQ | 8/6 | 20.28 |
| LTQ+MoDiff (Ours) | | **6.75** |
| LTQ | 8/4 | 297.21 |
| LTQ+MoDiff (Ours) | | **22.12** |

A line of research has focused on distilling diffusion models into few-step variants, which can achieve comparable generation quality within significantly fewer sampling steps. To evaluate the generalizability of MoDiff in this setting, we conduct experiments with MixDQ (Zhao et al., 2024), a method specifically designed for few-step diffusion. We use SDXL-Turbo as the backbone and apply 2, 4, and 8 sampling steps for image generation on the MS-COCO 2014 dataset (Lin et al., 2014), generating 10,000 images for FID computation. As shown in Table 17, our method is compatible with MixDQ and further improves performance in the few-step diffusion regime. The performance indicates that it is more challenging to lower the activation bit for SDXL-Turbo.

*Table 17.* FID on MS-COCO using SDXL-Turbo and MixDQ across different generation steps. The best performance is **bolded**.

| Step | Bits(W/A) | MixDQ | MixDQ+MoDiff |
|------|-----------|-------|--------------|
| 2 | 8/8 | 46.48 | **46.30** |
|   | 8/6 | 318.68 | **193.17** |
|   | 8/4 | 304.77 | **192.65** |
| 4 | 8/8 | **44.29** | 44.74 |
|   | 8/6 | 318.57 | **191.59** |
|   | 8/4 | 325.68 | **192.74** |
| 8 | 8/8 | 44.61 | **43.30** |
|   | 8/6 | 347.75 | **210.38** |
|   | 8/4 | 348.75 | **212.68** |

### D.5. Results on Warm-up

To verify that warm-up is not the primary source of improvement, we conduct an ablation study by applying warm-up to the baseline and removing it from MoDiff. The experiments are performed using the DDIM sampler on CIFAR-10 with LCQ. As shown in Table 18, MoDiff consistently outperforms the baseline under fair comparison, indicating that the observed performance gains are not attributable to the warm-up mechanism.

*Table 18.* FID on CIFAR-10 using the DDIM sampler in the ablation study of warm-up. The best performance is **bolded**.

| Bits (W/A) | LCQ w/o warmup | LCQ w/ warmup | LCQ+MoDiff w/o warmup | LCQ+MoDiff w/ warmup |
|------------|----------------|---------------|-----------------------|----------------------|
| 8/8 | 4.19 | **4.19** | 4.22 | 4.21 |
| 8/6 | 9.93 | 9.53 | 4.25 | **4.00** |
| 8/4 | 306.06 | 299.96 | 31.22 | **28.19** |

Moreover, as indicated by Theorem 4.4, warm-up can be achieved by repeatedly inputting $\mathbf{a}_T$. This process converges to the full-precision activation due to the contraction of the quantization error. As demonstrated in our experiments, approximately 4 to 5 steps are sufficient to reduce the quantization error to a negligible level on CIFAR-10 using 4-bit precision.

### D.6. Analysis on Memory Consumption

In the main paper, we present the trade-off analysis between computation cost and memory cost for MoDiff when generating a single image on CIFAR-10 with DDIM. In this section, we extend our analysis to larger batch sizes selected from $\{2, 4, 8\}$. The results are shown in Tables 19, 20, and 21. The results, shown in Tables 19, 20, and 21, demonstrate that MoDiff significantly reduces computation cost while incurring only a minimal increase in memory usage.

*Table 19.* The relationship between BOPs and memory usage of our method using DDIM on CIFAR-10 for generation with batch size 2.

| Measurement | W8A2 | W8A4 | W8A8 | W8A32 |
|-------------|------|------|------|-------|
| GBops | 204 | 410 | 918 | 3272 |
| Memory (Mb) | 36.49 | 38.89 | 43.69 | 36.09 |

*Table 20.* The relationship between BOPs and memory usage of our method using DDIM on CIFAR-10 for generation with batch size 4.

| Measurement | W8A2 | W8A4 | W8A8 | W8A32 |
|-------------|------|------|------|-------|
| GBops | 408 | 820 | 1836 | 6544 |
| Memory (Mb) | 38.89 | 43.69 | 53.28 | 38.09 |

*Table 21.* The relationship between BOPs and memory usage of our method using DDIM on CIFAR-10 for generation with batch size 8.

| Measurement | W8A2 | W8A4 | W8A8 | W8A32 |
|---|---|---|---|---|
| GBops | 906 | 1640 | 3672 | 13088 |
| Memory (Mb) | 43.69 | 53.28 | 72.47 | 42.09 |

## E. Compared to PTQD

Post-Training Quantization for Diffusion Models (PTQD) aims to reduce quantization error by post-processing quantized models, sharing a similar objective with our work. In this section, we highlight the key differences between MoDiff and PTQD. Compared to PTQD, MoDiff is (1) more general and flexible, (2) free from strong assumptions about error distribution, and (3) significantly more effective in low-precision scenarios.

(1) PTQD requires solver-specific adaptations to address variance and bias, while MoDiff can be applied across solvers without modification. Moreover, PTQD is restricted to standard diffusion models, whereas MoDiff also supports cached diffusion models by compensating for reuse errors in cached components.

(2) PTQD relies on strong assumptions about error distribution, specifically that quantization errors follow a Gaussian distribution after input rescaling. This assumption can introduce inaccuracies in error estimation. In contrast, MoDiff leverages the widely observed similarity between timesteps, which is well-supported by prior works (Ma et al., 2024b).

(3) MoDiff performs well in low-precision activation settings, whereas PTQD fails entirely. To demonstrate this, we evaluate both methods on CIFAR-10 with W8A4 quantization. PTQD yields an FID of 397.12 and fails to produce meaningful images, while MoDiff achieves a much lower FID of 13.41.

## F. Comprehensive Visualization Results

In this section, we present visualization results for CIFAR-10, LSUN-Churches, LSUN-Bedroom, and MS-COCO-2014. These results illustrate the performance that MoDiff can achieve. For instance, as shown in Figure 5, LCQ+MoDiff closely aligns with full-precision generation at W8A4, whereas LCQ only captures the image textures. Additionally, LCQ+MoDiff can still generate recognizable images at W8A3, albeit with some loss of detail.

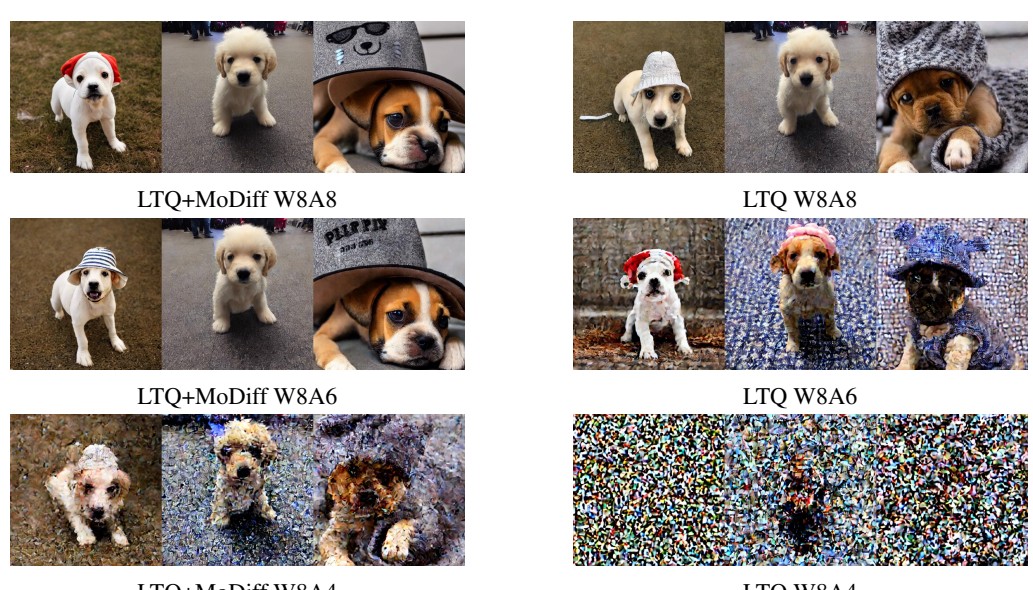

LTQ+MoDiff W8A8      LTQ W8A8

LTQ+MoDiff W8A6      LTQ W8A6

LTQ+MoDiff W8A4      LTQ W8A4

*Figure 4.* Visualization of MS-COCO-2014 generated using LTQ and LTQ+MoDiff under 8-bit weight quantization precisions on Stable Diffusion v1.4.

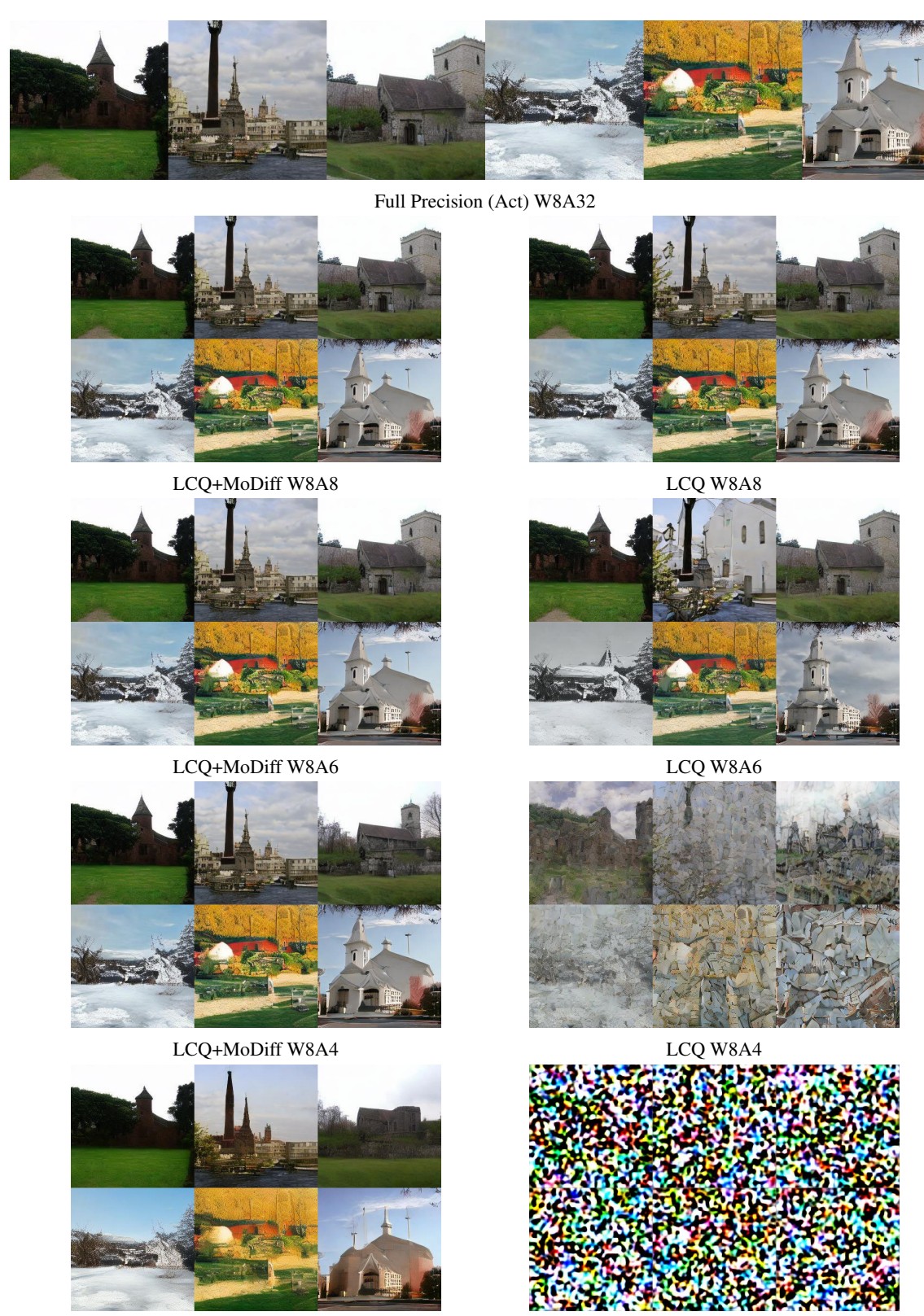

Full Precision (Act) W8A32

LCQ+MoDiff W8A8 · LCQ W8A8

LCQ+MoDiff W8A6 · LCQ W8A6

LCQ+MoDiff W8A4 · LCQ W8A4

LCQ+MoDiff W8A3 · LCQ W8A3

*Figure 5.* Visualization of LSUN-Churches $256 \times 256$ generated using LCQ and LCQ+MoDiff under 8-bit weight quantization precisions.

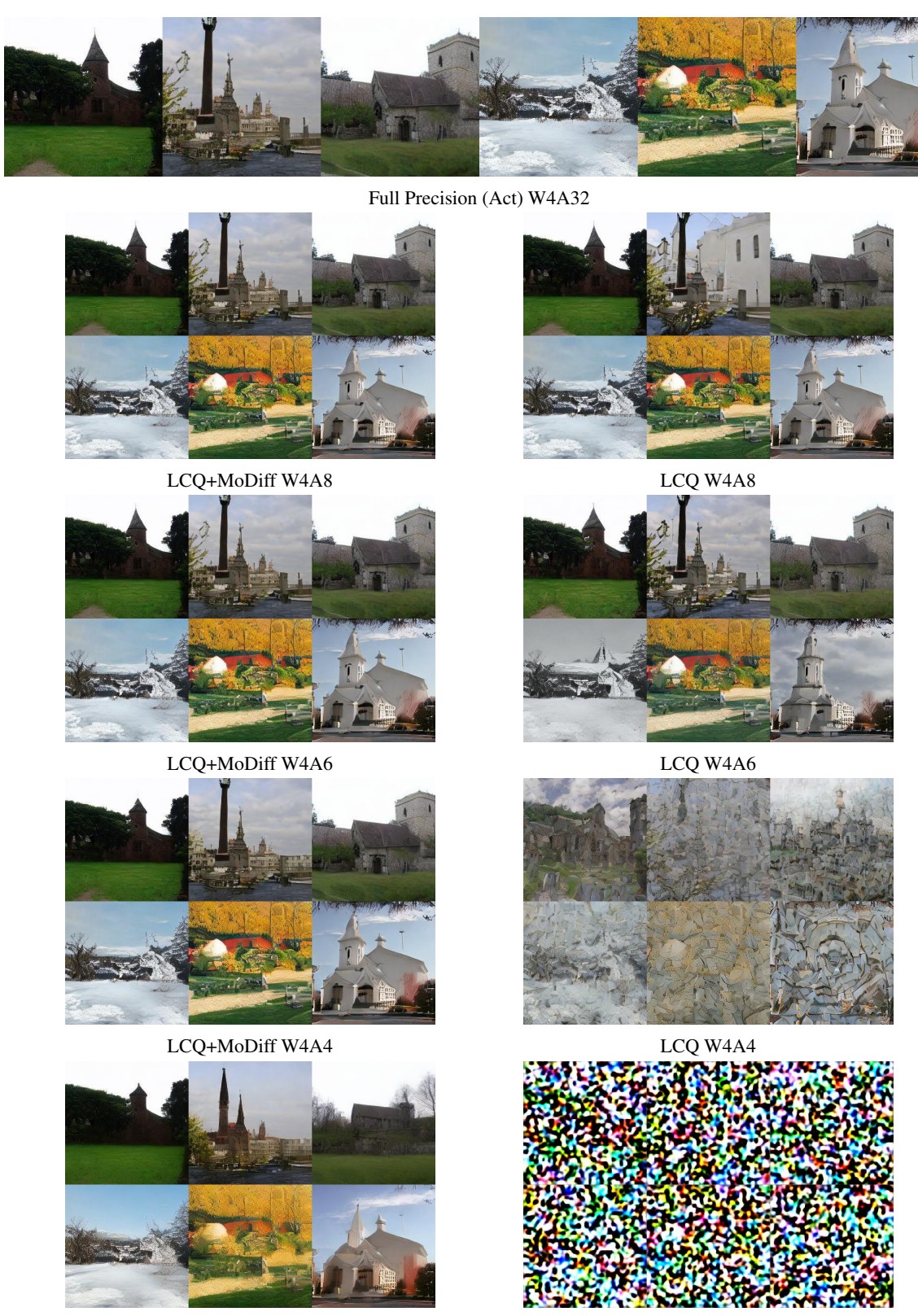

*Figure 6.* Visualization of LSUN-Churches $256 \times 256$ generated using LCQ and LCQ+MoDiff under 4-bit weight quantization precisions.

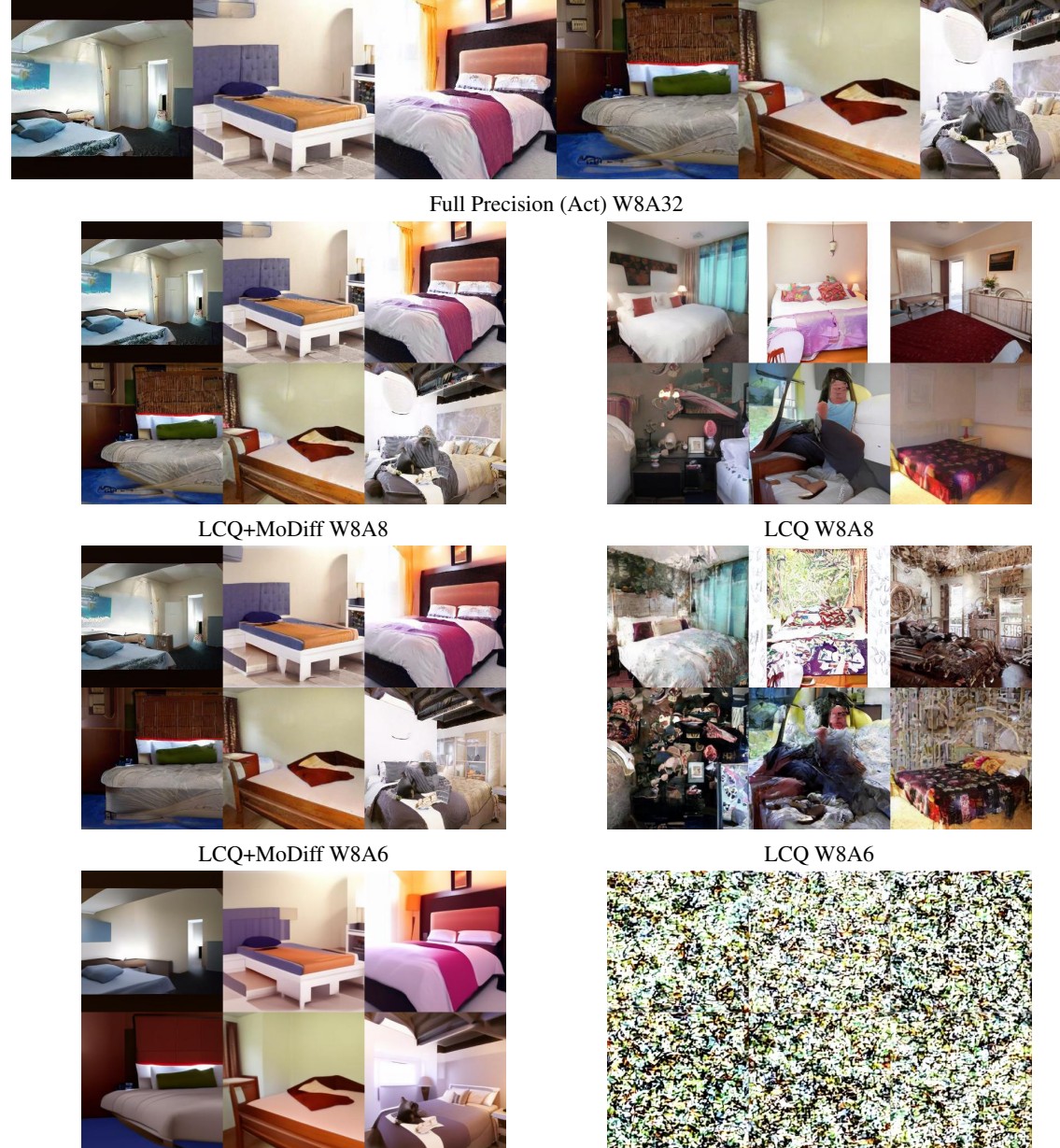

Figure 7. Visualization of LSUN-bedrooms $256 \times 256$ generated using LCQ and LCQ+MoDiff under 8-bit weight quantization precisions.

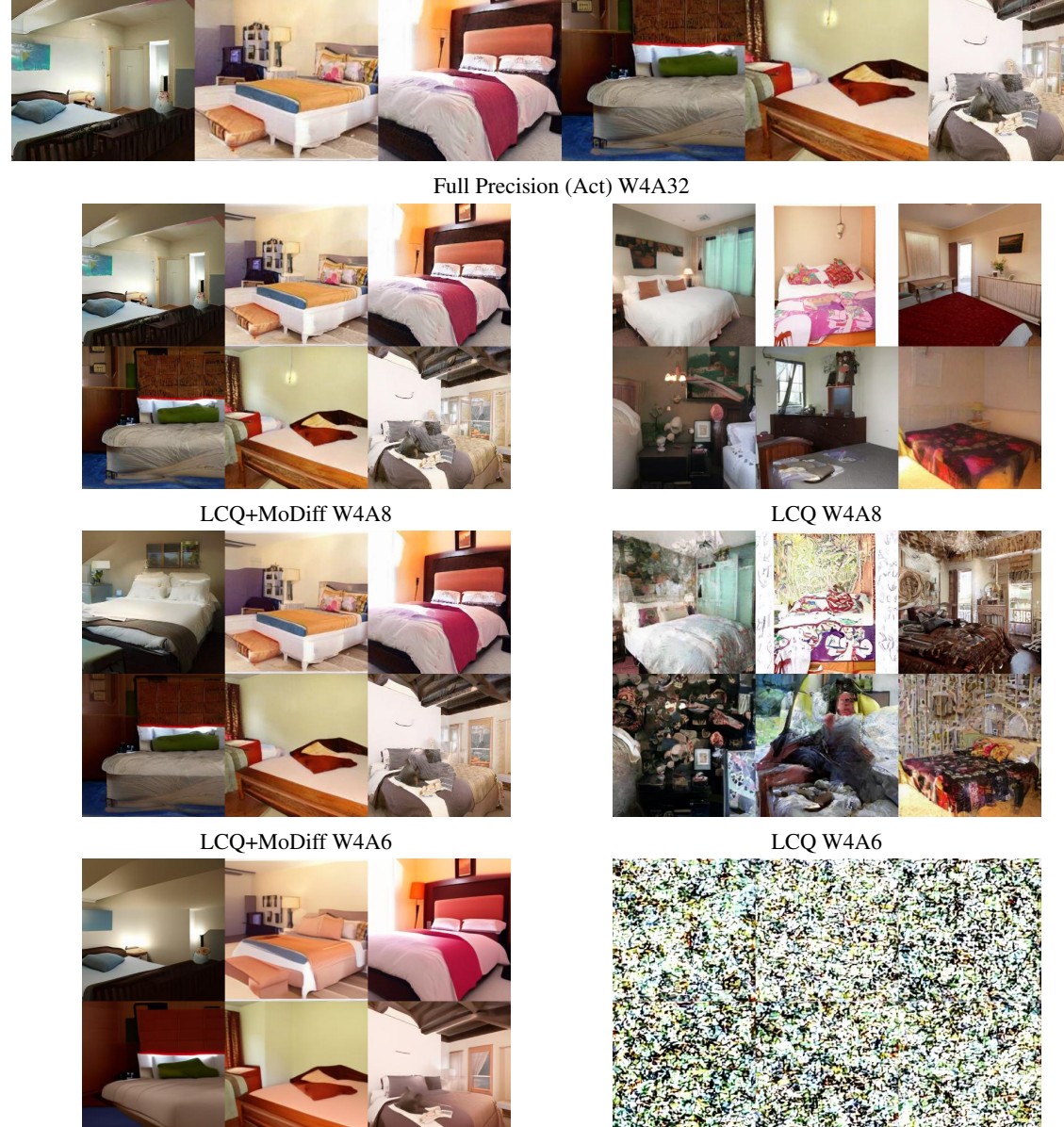

Figure 8. Visualization of LSUN-bedrooms $256 \times 256$ generated using LCQ and LCQ+MoDiff under 4-bit weight quantization precisions.

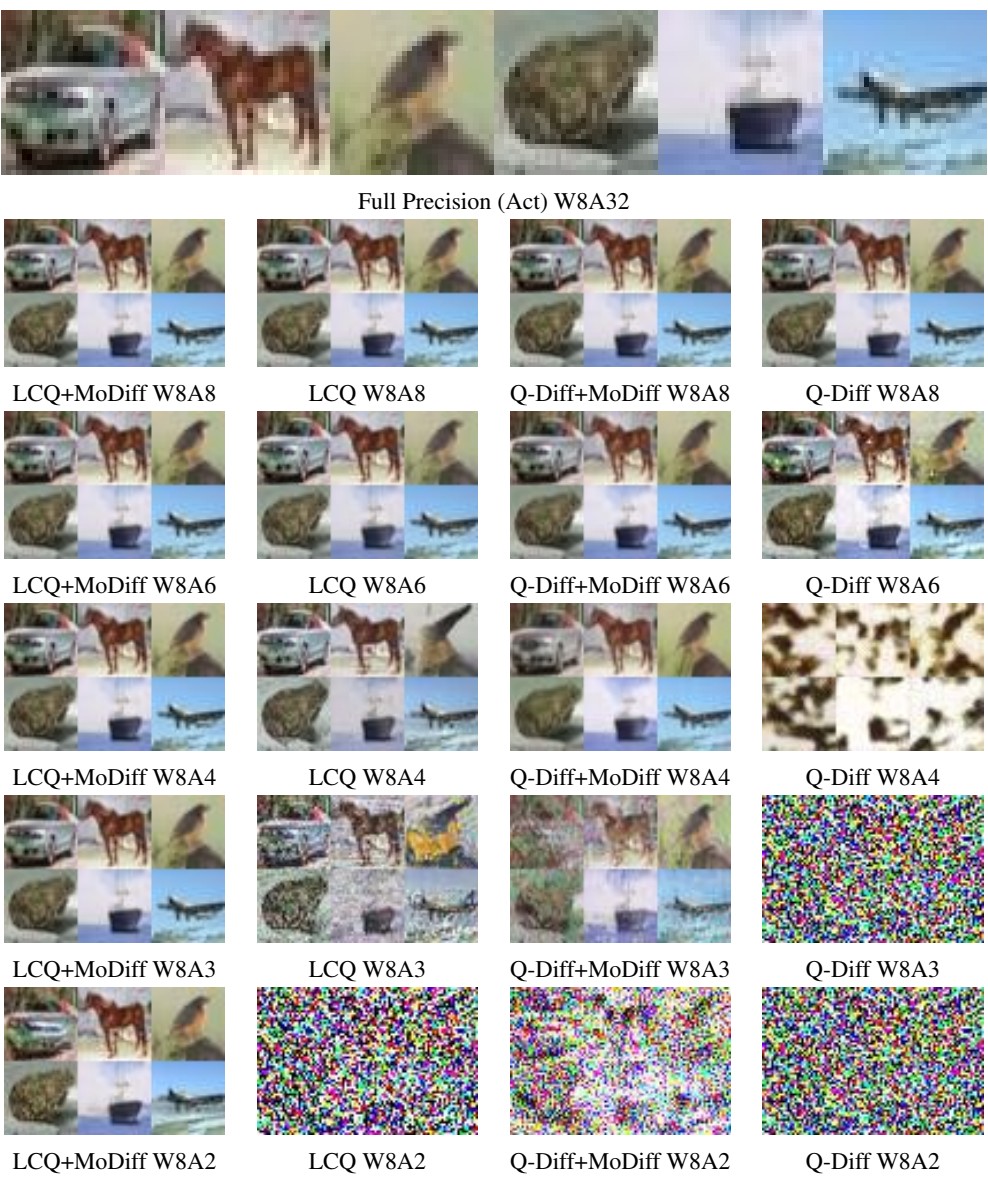

Figure 9. Visualization of LSUN-bedrooms $256 \times 256$ generated using LCQ, LCQ+MoDiff, Q-Diff, and Q-Diff+MoDiff under 8-bit weight quantization precisions.

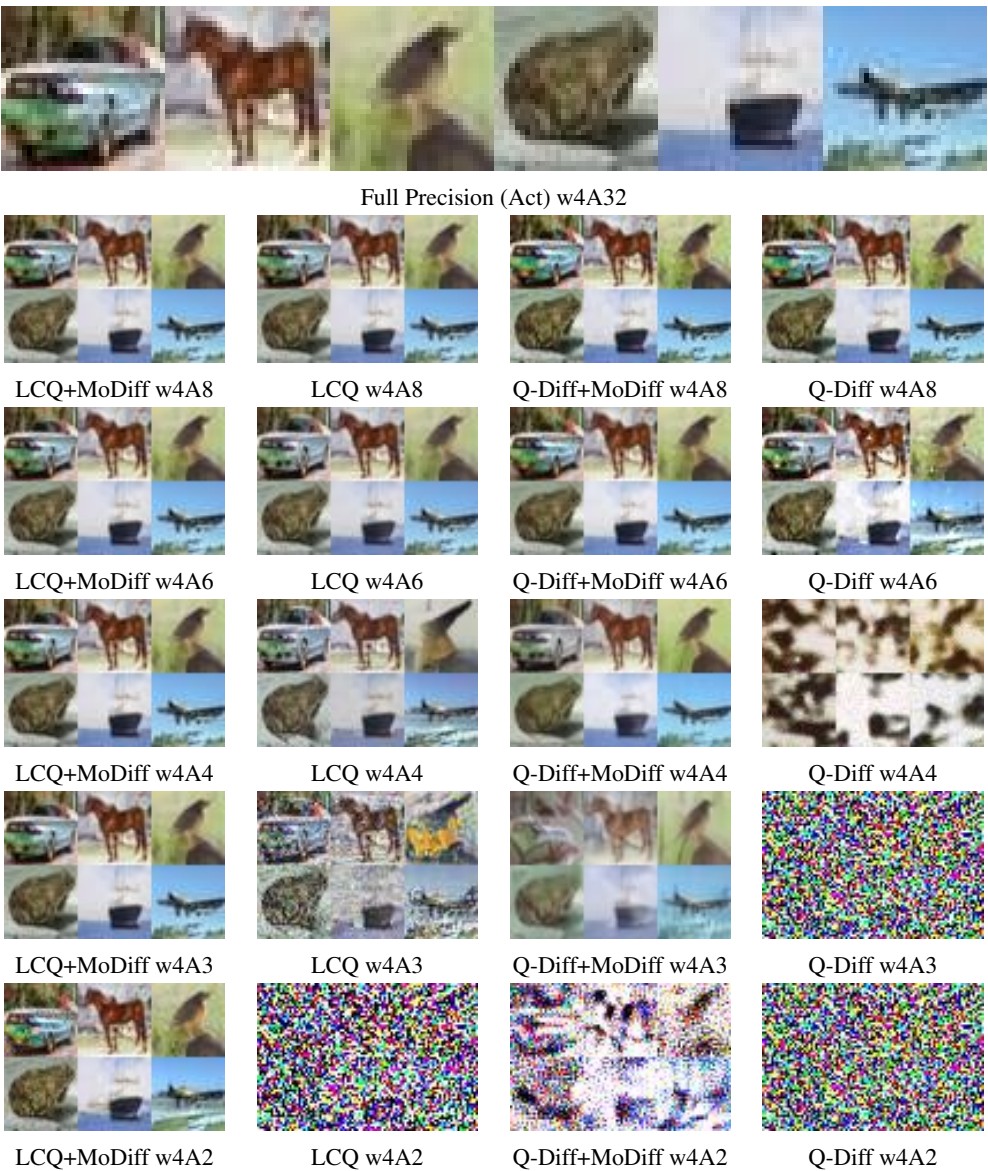

*Figure 10.* Visualization of LSUN-bedrooms $256 \times 256$ generated using LCQ, LCQ+MoDiff, Q-Diff, and Q-Diff+MoDiff under 4-bit weight quantization precisions.

