# OpenReview forum: "Modulated Diffusion:  Accelerating Generative Modeling with Modulated Quantization"
_ICML.cc/2025/Conference — ICML 2025 poster_

### Official Review · Reviewer_YCE9 · 2025-02-27

**Overall Recommendation:** 1

**Summary:**

This work introduces MoDiff, a novel framework for accelerating diffusion models by combining modulated quantization and error compensation. It enhances existing techniques like caching and quantization, offering a more efficient approach without sacrificing generation quality. MoDiff reduces activation quantization from 8 bits to 3 bits in post-training quantization (PTQ) on datasets like CIFAR-10 and LSUN, with no performance loss.

**Claims And Evidence:**

N/A

**Essential References Not Discussed:**

N/A

**Experimental Designs Or Analyses:**

1. The superior results in the paper are mainly based on LCQ (i.e., dynamic channel-wise activation quantization). I believe that this scenario can not obtain any speedup and is never presented in any CNN quantization works. Moreover, the bit-width settings in this paper are a little bit wired (activation << weight). All these need to be further discussed.

2. w8a8 Q-Diffusion w/o MoDiff in some scenarios outperforms MoDiff, which needs to be further explained.

3. Ablation studies for only employing standard modulate quantization vs. baseline are missing.

4. Experiments related to Q-Diffusion may be not fair. For example, the implemental details in Sec. B are all inconsistent with Q-Diffusion.

**Methods And Evaluation Criteria:**

The models and datasets used in this paper are too old. For example, this paper's latest model LDM [1] was released in 2022 and most of experiments are based on CIFAR 32$\times$32. More advanced models (e.g., PixArt [2] or FLUX [3] ), metrics (e.g., Image Reward [4] or GenVal [5]), and datasets (e.g., DCI [6] or MJHQ [7]) should be involved.

[1] Rombach R, Blattmann A, Lorenz D, et al. High-resolution image synthesis with latent diffusion models[C]//Proceedings of the IEEE/CVF conference on computer vision and pattern recognition. 2022: 10684-10695.

[2] Chen J, Yu J, Ge C, et al. Pixart-
: Fast training of diffusion transformer for photorealistic text-to-image synthesis[J]. arXiv preprint arXiv:2310.00426, 2023.

[3] https://github.com/black-forest-labs/flux

[4] Xu J, Liu X, Wu Y, et al. Imagereward: Learning and evaluating human preferences for text-to-image generation[J]. Advances in Neural Information Processing Systems, 2024, 36.

[5] Ghosh D, Hajishirzi H, Schmidt L. Geneval: An object-focused framework for evaluating text-to-image alignment[J]. Advances in Neural Information Processing Systems, 2024, 36.

[6] Urbanek J, Bordes F, Astolfi P, et al. A picture is worth more than 77 text tokens: Evaluating clip-style models on dense captions[C]//Proceedings of the IEEE/CVF Conference on Computer Vision and Pattern Recognition. 2024: 26700-26709.

[7] Li D, Kamko A, Akhgari E, et al. Playground v2. 5: Three insights towards enhancing aesthetic quality in text-to-image generation[J]. arXiv preprint arXiv:2402.17245, 2024.

**Other Comments Or Suggestions:**

Eq. (18) needs to be fixed.

**Other Strengths And Weaknesses:**

Strengths:

 The idea is novel and interesting.

Weaknesses:

Some parts of this paper are hard to read. For example, the expression in L237-260 and L324-329 is very confusing and difficult to understand.

**Questions For Authors:**

This paper currently focuses on diffusion with hundreds of denoising steps during inference. However, few-step diffusion models are widely used. Can this work achieve performance improvement when being applied to these models? As shown in Eq. (30) and (31), the errors are the same w/ and w/o error compensation for a 2-step diffusion model.

**Relation To Broader Scientific Literature:**

This paper lacks a lot of advanced baselines, e.g., MixDQ and EfficientDM.  It is better to compare and combine this work with them to show the contribution.

[8] Zhao T, Ning X, Fang T, et al. Mixdq: Memory-efficient few-step text-to-image diffusion models with metric-decoupled mixed precision quantization[C]//European Conference on Computer Vision. Cham: Springer Nature Switzerland, 2024: 285-302.

[9] He Y, Liu J, Wu W, et al. Efficientdm: Efficient quantization-aware fine-tuning of low-bit diffusion models[J]. arXiv preprint arXiv:2310.03270, 2023.

**Theoretical Claims:**

1. The assumption in Eq. (29) requires proper bit-width choices, which are constrained by the tensor shape as illustrated in Eq. (78). I suggest the authors discuss the feasible choices for different linear operations in diffusion.

2. The inputs in Eq. (25) and Eq. (28) are not quantized. Does this mean that the authors use the full-precision denoising at $T$?

---

> ### Author Rebuttal · Authors · 2025-04-01
>
> Thanks for recognizing the novelty of our paper. We believe there are some misunderstandings about our implementation, and we will address your questions with the following experiments.
>
> **Methods And Evaluation Criteria: Q1**
>
> MoDiff focuses on quantization rather than the latest models or datasets. To clarify its generality, we evaluate MoDiff on Stable Diffusion with MS-COCO and DiT on ImageNet using public quantized checkpoints [1,2]. Results show that MoDiff generalizes well across architectures and datasets.
>
> | W/A | Baseline | MoDiff |
> |:-:|--:|--:|
> | 8/8 | 54.80 | 53.76 |
> | 8/6 | 200.26 | 54.74 |
> | 8/4 | 271.87 | 90.91 |
>
> Table 1: Stable Diffusion
>
> | W/A | Baseline | MoDiff |
> |:-:|--:|--:|
> | 8/8 | 54.80 | 53.76 |
> | 8/6 | 200.26 | 54.74 |
> | 8/4 | 271.87 | 90.91 |
>
> Table 2: DiT
>
> [1] Q-Diffusion. ICCV 2023.
>
> [2] PTQ4DiT. NeurIPS 2024.
>
> **Theoretical Claims: Q1**
>
> The corollary validates our assumptions and guides design; it's not a strict per-layer rule nor the tightest bound. We aim to extend this to mixed-precision in future work.
>
> **Theoretical Claims: Q2**
>
> We warm up by running the first timestep in full precision and then using low-bit activations. This adds negligible overhead. We apply warm-up to baselines for fairness, showing the performance gain is not from warm-up.
>
> | W/A | Baseline + Warmup | MoDiff |
> |:-:|--:|--:|
> | 8/8 | 4.19 | 4.21 |
> | 8/6 | 9.53 | 4.28 |
> | 8/4 | 299.96 | 28.19 |
>
> **Experimental Designs Or Analyses: Q1**
>
> First, MoDiff is quantization-method agnostic. We follow [1]'s channel-wise dynamic quantization (hardware-unfriendly, but illustrative), and also test Q-Diffusion and tensor-wise quantization (Table 1, 11), showing consistent FID improvements.
>
> Second, low-bit activation cases are rare due to the large quantization error they introduce. MoDiff breaks the limitations of existing methods by effectively reducing activation quantization error. Similar configurations are also used in prior works [2,3].
>
> [1] DGQ. ICLR 2025.
>
> [2] PikeLPN. CVPR 2024.
>
> [3] WRPN. ICLR 2018.
>
> **Experimental Designs Or Analyses: Q2**
>
> The baseline does not necessarily outperform MoDiff since MoDiff (1) has smaller quantization errors and (2) has better performance on sFID. Specifically, MoDiff closely matches the full-precision model across all metrics compared to the baseline, meaning its output is nearly the same as the ground truth. Moreover, MoDiff outperforms the baseline in sFID. It is inaccurate to conclude that the baseline is better considering all metrics.
>
> **Experimental Designs Or Analyses: Q3**
>
> To address your concern, we show how error compensation improves sFID for CIFAR-10 with DDIM.
>
> | W/A | Baseline | w/o Compensation | w/ Compensation |
> |:-:|--:|--:|--:|
> | 8/6 | 4.61 | 4.41 | 4.40 |
> | 8/4 | 13.07 | 10.21 | 4.38 |
> | 8/3 | 33.97 | 25.42 | 4.38 |
>
> **Experimental Designs Or Analyses: Q4**
>
> Our comparison with Q-Diffusion is fair. We only make minimal changes and necessary adaptations to integrate our method. Specifically, we (1) apply modulated computation and remove bias for correctness, (2) use temporal differences instead of raw activations for calibration since they serve as inputs during sampling. The last implementation is (3) omitting block-wise quantization for training stability. To demonstrate that (3) does not account for MoDiff’s performance gains, we applied it to Q-Diffusion with W8A8 on CIFAR-10, yielding FID 4.76 (worse than baseline). Our implementation also reproduces Q-Diffusion results without modulation, confirming consistency.
>
> **Relation To Broader Scientific Literature: Q1**
>
> We exclude MixDQ due to a lack of public code. EfficientDM uses quantization-aware training, which differs from our post-training quantization focus.
>
> **Other Strengths And Weaknesses: Q1**
>
> Lines 237–260 explain how errors accumulate in modulated computation and how our compensation method addresses this. Specifically, $\hat a\_t$ represents the activation computed in the previous timestep, while $a\_t$ is the desired (unquantized) activation. Modulated computation reuses $A(\hat a\_t)$ instead of $A(a\_t)$, so computing temporal difference based on $a_t$ in the next step will introduce errors. Our compensation technique corrects this by computing temporal difference based on $\hat a_t$.
>
> Lines 324–329 emphasize that compensation is more critical in practice than theory suggests. In practice, diffusion models suffer from significant error accumulation as quantization errors build up layer by layer. The severity of error accumulation highlights the necessity of applying error compensation.
>
> **Other Comments Or Suggestions: Q1**
>
> Thanks for pointing out the missing right parenthesis. We will correct the typo in the revised version.
>
> **Questions For Authors: Q1**
>
> We run tensor-wise quantization on CIFAR-10 with 20-step DDIM. MoDiff consistently improves low-bit quality.
>
> | W/A | Baseline | MoDiff |
> |:-:|--:|--:|
> | 8/8 | 6.93 | 6.90 |
> | 8/6 | 20.28 | 6.75 |
> | 8/4 | 297.21 | 22.12 |

---

> > ### Comment · Reviewer_YCE9 · 2025-04-01
> >
> > Thanks for the reply. I still have the following concerns:
> >
> > 1. The results on Stable-Diffusion and DiT are not sufficiently convincing. I believe that including the more advanced models, datasets, and metrics, as I mentioned in the first round, is necessary to validate the applicability in practice.
> > 2. The warm-up and quantized denosing require people to save two models (e.g., full-precision and quantized version), which will bring too much memory consumption and eliminate one of the important effects of quantization (i.e., reducing memory consumption).
> > 3. Since there are no kernel implementations for fast $\texttt{high-bit weight} \times \texttt{low-bit activation}$ (they should have the same bit-width in most cases for real-time speedup in current processors), the paper’s approach lacks practicality (I believe this is not a theoretical paper).
> > 4. The author does not answer my question, where I mentioned that for w8a8 in some datasets, instead of the most cases. I merely hope the author to analyze these special cases.
> > 5. [The official code of MixDQ](https://github.com/A-suozhang/MixDQ) was made available online last year. Therefore, "lack of public code" is unreasonable. Moreover, I think EfficientDM should be included for comparison, since it is more efficient than the reconstruction-based PTQ baseline used in the paper. EfficientDM finetunes the LoRA branch, which brings even smaller training overhead than the adaptive rounding in the reconstruction. Thus, I think EfficientDM is very special, which does not train the model weights like the norm QAT, and should also be included. At least, I think the author should combine their methods with more advanced baselines. Q-Diffusion used in the paper is very slow, and thus hard to apply to large-scale models in practice.
> > 5. The author does not include the results for the few-step diffusion models. I think 20 steps is a moderate-step scenario. I suggest the author use step-distilled models.
> >
> > Overall, the contribution of the paper in practice is insufficient, and I decide to further lower the score.

---

> > > ### Author Response · Authors · 2025-04-04
> > >
> > > Thank you for your detailed feedback. However, these are misunderstandings of our method and response. To solve the concerns, we will clarify the points of confusion and provide additional experiments.
> > >
> > > **Q1. Including the more advanced models, datasets, and metrics is necessary.**
> > >
> > > We would like to emphasize three key points: (1) our method is general, and we demonstrate its effectiveness on widely recognized models; (2) our comparisons are both up-to-date and fair; and (3) we conducted additional experiments on SDXL-Turbo, as suggested, to further address your concerns. To summarize, our method shows consistent improvement in all experiments and ablation studies, which already clearly validates the scientific merits of this work.
> > >
> > > Given the rapid development of generative models, it is neither feasible nor necessary to always chase the latest models and datasets. In this context, Stable Diffusion, DiT, and their associated datasets and metrics are still widely used in recent quantization works and remain representative and foundational [1,2,3,4]. Therefore, our comparisons on these architectures are fair.
> > >
> > > [1] PTQ4DiT. NeurIPS 2024.
> > >
> > > [2] BiDM. NeurIPS 2024.
> > >
> > > [3] StepbaQ. NeurIPS 2024.
> > >
> > > [4] DGQ. ICLR 2025.
> > >
> > > **Q2. The warm-up and quantized denoising require people to save two models.**
> > >
> > > This is a misunderstanding of our method. We save only one quantized model; full precision is used solely for activations in the first time step. For subsequent steps, the same model is used with modulated quantization applied. Equation (6) in our main paper clearly illustrates this core idea of modulated computing. Additionally, we conduct experiments showing that removing the warm-up phase can still significantly outperform the baselines.
> > >
> > > |Bits (W/A)|Baseline|MoDiff w/o Warm|MoDiff w/ Warm|
> > > |-|-|-|-|
> > > | 8/8| 4.19| 4.22|4.21|
> > > | 8/6 | 9.93| 4.25|4.00|
> > > | 8/4| 28.19| 31.22| 28.19|
> > >
> > > **Q3. There are no kernel implementations on this setting.**
> > >
> > > We would like to emphasize three points: (1) hardware is not the focus of our work; (2) MoDiff represents a leap toward enabling low-bit activations; and (3) similar experimental settings are commonly used in existing works.
> > >
> > > First, while MoDiff is not hardware-centric, our experimental design demonstrates its effectiveness. Moreover, lowering the weight bandwidth to match the suggested setting is feasible, as shown in our W4A4 results.
> > >
> > > Second, this research is forward-looking instead of being restricted by existing hardware. MoDiff is a leap toward low-bit activation. This line of research has the potential of eventually reducing to 1 bit, which further influences how future hardware should be designed (such as logic-based operations).
> > >
> > > Finally, similar settings are adopted in very recent works on this topic [1,2,3] to validate the effectiveness of their methods.
> > >
> > > [1] DGQ. ICLR 2025.
> > >
> > > [2] PikeLPN. CVPR 2024.
> > >
> > > [3] WRPN. ICLR 2018.
> > >
> > > **Q4. I mentioned that for w8a8 in some datasets, instead of the most cases.**
> > >
> > > The case you mention is W8A8 in Q-Diffusion, where the baseline achieves a slightly better FID than MoDiff (3.75 vs. 4.10), but MoDiff achieves a better sFID (4.49 vs. 4.39). First, these small differences do not conclusively indicate that Q-Diffusion has better generation quality, as MoDiff also outperforms in sFID. Second, MoDiff exhibits smaller quantization error, better aligning with the full-precision model (FID 4.24, sFID 4.41). As shown in Table 1 (W8A8 case) and Table 3 (W8A8 case) of [1], minor quantization error can slightly reduce FID, which aligns with our findings. In a nutshell, the results in W8A8 are reasonable.
> > >
> > > [1] Q-Diffusion. CVPR 2023.
> > >
> > > **Q5. Experiments on MixDQ and Few-step diffusion models.**
> > >
> > > To address your concern, we conducted additional experiments on 2,4,8-step SDXL-Turbo using MixDQ as the baseline. We generated 10,000 images to compute FID, and the results demonstrate that our method is compatible with both MixDQ and few-step diffusion models, further improving their performance. The implementation is available in the following anonymous GitHub repository: https://anonymous.4open.science/r/MixDQ-MoDiff-7C52/
> > >
> > > 2-Step
> > > |W/A|MixDQ|MixDQ+MoDiff|
> > > |-|-|-|
> > > |8/8|46.48|**46.30**|
> > > |8/6|318.68|**193.17**|
> > > |8/4|304.77|**192.65**|
> > >
> > > 4-Step
> > > |W/A|MixDQ|MixDQ+MoDiff|
> > > |-|-|-|
> > > |8/8|44.29|**44.74**|
> > > |8/6|318.57|**191.59**|
> > > |8/4|325.68|**192.74**|
> > >
> > > 8-Step
> > > |W/A|MixDQ|MixDQ+MoDiff|
> > > |-|-|-|
> > > |8/8|44.61|**43.30**|
> > > |8/6|347.75|**210.38**|
> > > |8/4|348.75|**212.68**|
> > >
> > > **Q6. Combine with EfficientDM**
> > >
> > > First, MoDiff is a post-processing framework, and how to integrate it into the training process is unclear, which we consider as an important future work.
> > > Second, although EfficientDM is highly efficient, it alters model parameters, rendering it an unsuitable baseline for direct comparison with PTQ methods. As noted in previous studies (very recent ones) [1,2], EfficientDM is typically excluded when evaluating PTQ approaches.
> > >
> > > [1] DGQ. ICLR 2025.
> > >
> > > [2] StepbaQ. NeurIPS 2024.

---

### Official Review · Reviewer_wYCA · 2025-03-11

**Overall Recommendation:** 3

**Summary:**

The paper investigates the shortcomings of current acceleration methods for diffusion models, such as caching and quantization, which suffer from error accumulation and high approximation errors, and introduces MoDiff—a novel framework that accelerates diffusion models through modulated quantization combined with error compensation. The authors support their proposal with theoretical analyses that detail quantization errors and the benefits of the error compensation mechanism, and they validate their method through experiments on datasets like CIFAR-10, LSUN-Churches, and LSUN-Bedroom, demonstrating that activation precision can be reduced from 8 bits to as low as 3 bits without any performance degradation in a training-free manner.

**Claims And Evidence:**

Yes.

Figure 1 offers an interesting analysis that clearly motivates the use of modulated quantization, while the experimental results robustly validate the authors’ claims by demonstrating that MoDiff works well on various diffusion weight and activation precision.

**Essential References Not Discussed:**

There are some caching methods missing discussion. The latter two are about video caching but it shares some similar ideas to image caching.

- DiTFastAttn: Attention Compression for Diffusion Transformer Models, Neurips2024
- FasterCache: Training-Free Video Diffusion Model Acceleration with High Quality, ICLR2025
- Adaptive Caching for Faster Video DiTs, arxiv.

**Experimental Designs Or Analyses:**

The experiments are generally convincing, but they have minor issues.

- All experiments were conducted on small-scale datasets such as CIFAR-10, LSUN-Churches, and LSUN-Bedroom, whereas it is standard practice to validate a research idea on larger datasets like ImageNet—especially since the proposed methods are training-free, making scalability feasible.
- Additionally, section 5.3 should extend its evaluation to include more recent samplers, such as flow models, EDM, and DPM-solver, as relying solely on DDIM and DDPM is insufficient for the SOTA validation.
- The ablation study on error compensation would also benefit from employing direct image quality metrics like FID, IS, or P&R instead of the L2 metric, which is less convincing for assessing image quality.
- Table 1 contains an error in the bolding; in the 8/6 bits quantization setting, LCQ has a lower FID compared to LCQ+MoDiff, indicating an error in the highlighted results.

**Methods And Evaluation Criteria:**

Yes. The method part is well-motivated.

**Other Comments Or Suggestions:**

No any other suggestions.

**Other Strengths And Weaknesses:**

please see above.

**Questions For Authors:**

No any other questions.

**Relation To Broader Scientific Literature:**

This paper examines the limitations of current caching techniques and post-training quantization (PTQ) methods, and based on these insights, the authors introduce MoDiff. Notable related approaches include DeepCache: Accelerating Diffusion Models for Free (NeurIPS 2024) and Q-Diffusion: Quantizing Diffusion Models (ICCV2023).

**Theoretical Claims:**

Yes. From my point of view, the proofs in section 4.2 and 4.3 are correct and easy to understand.

---

> ### Author Rebuttal · Authors · 2025-04-01
>
> Thank you for recognizing the novelty, effectiveness, and clarity of our paper. We are glad to address your questions.
>
> **Experimental Designs Or Analyses: Q1. All experiments were conducted on small-scale datasets such as CIFAR-10, LSUN-Churches, and LSUN-Bedroom, whereas it is standard practice to validate a research idea on larger datasets like ImageNet—especially since the proposed methods are training-free, making scalability feasible.**
>
> To address your concern, we conduct experiments following [1], using DiT on ImageNet 256×256 with tensor-wise dynamic quantization. The results demonstrate that our method consistently improves generation quality at low activation bit widths:
>
> | W/A | Baseline | MoDiff |
> |:-:|--:|--:|
> | 8/8 | 54.80 | **53.76** |
> | 8/6 | 200.26 | **54.74** |
> | 8/4 | 271.87 | **90.91** |
>
> [1] PTQ4DiT: Post-training Quantization for Diffusion Transformers. NeurIPS 2024.
>
> **Experimental Designs Or Analyses: Q2. Should extend its evaluation to include more recent samplers, such as flow models, EDM, and DPM-solver.**
>
> To address your concern, we perform tensor-wise quantization using the DPM-Solver-2 on CIFAR-10 with 20 sampling steps. Additionally, we conduct experiments with the PLMS solver using 50 steps on Stable Diffusion with MS-COCO-2014. In both cases, MoDiff consistently improves FID scores across different solvers, even with reduced sampling steps.
>
> | W/A | Baseline | MoDiff |
> |:-:|--:|--:|
> | 8/8 | 3.92 | **3.91** |
> | 8/6 | 10.82 | **3.92** |
> | 8/4 | 299.72 | **26.54** |
>
> Table1: DPM on CIFAR-10
>
> | W/A | Baseline | MoDiff |
> |:-:|--:|--:|
> | 8/8 | 54.80 | **53.76** |
> | 8/6 | 200.26 | **54.74** |
> | 8/4 | 271.87 | **90.91** |
>
> Table 2: PLMS on MS-COCO
>
> **Experimental Designs Or Analyses: Q3. The ablation study on error compensation would also benefit from employing direct image quality metrics like FID, IS, or P\&R instead of the L2 metric, which is less convincing for assessing image quality.**
>
> To address your concern, we conduct experiments to show the sFID measurement with or without error compensation on CIFAR10 with DDIM. The results show that error compensation reduces the error accumulation in low bit activation cases.
>
> | W/A | Baseline | w/o Compensation | w/ Compensation |
> |:-:|--:|--:|--:|
> | 8/6 | 4.61 | 4.41 | **4.40** |
> | 8/4 | 13.07 | 10.21 | **4.38** |
> | 8/3 | 33.97 | 25.42 | **4.38** |
>
> **Experimental Designs Or Analyses: Q4. Table 1 contains an error in the bolding.**
>
> Thank you for pointing out the incorrect bold formatting. We will revise it in a new version.
>
> **Essential References Not Discussed: Q1. There are some caching methods missing discussion.**
>
> Thank you for mentioning the relevant papers. We will include them in the related works section of the revised version.

---

### Official Review · Reviewer_32oq · 2025-03-12

**Overall Recommendation:** 2

**Summary:**

The author introduces MoDiff, a framework designed to accelerate generative modeling by addressing challenges in caching and post-training quantization (PTQ). MoDiff incorporates modulated quantization and error compensation to reduce quantization errors and mitigate error accumulation. Theoretical analysis supports its effectiveness, and experiments on CIFAR-10 and LSUN show that MoDiff enables PTQ to operate at as low as 3-bit activation quantization.

**Claims And Evidence:**

The submission claims that existing caching methods introduce significant computation errors, but leveraging temporal stability in activation patterns can mitigate these issues. This claim is supported by empirical observations and visualizations (Figure 1), which show that temporal differences in activations have a more stable and concentrated distribution, reducing outliers and error accumulation.

**Essential References Not Discussed:**

MoDiff introduces a novel quantization approach that is closely related to existing post-training quantization methods like PTQD. PTQD addresses the challenges of quantization noise and its accumulation by introducing bias correction and variance schedule calibration. Similarly, MoDiff aims to mitigate quantization error and error accumulation. A direct comparison with PTQD would help clarify the advantages and distinctions of MoDiff.

[1] PTQD: Accurate Post-Training Quantization for Diffusion Models.

**Experimental Designs Or Analyses:**

The experimental design and analysis are well-structured, assessing MoDiff's effectiveness across multiple datasets (CIFAR-10, LSUN) using standard evaluation metrics (IS, FID, sFID). The results show that MoDiff preserves generation quality while significantly reducing computational costs.

**Methods And Evaluation Criteria:**

The proposed methods and evaluation criteria are aligned with the problem. MoDiff is tested on relevant benchmark datasets (CIFAR-10, LSUN) using widely accepted evaluation metrics like FID, IS, and SFID. The study also includes various quantization methods and bit-width settings for comprehensive comparison.

**Other Comments Or Suggestions:**

There is a missing right parenthesis in Equation (18).

**Other Strengths And Weaknesses:**

1、The paper lacks experiments on larger models, such as Stable Diffusion, which would better demonstrate its scalability and real-world applicability.

2、MoDiff's use of modulated quantization and error compensation is highly restrictive, as it can only be applied to linear modules. This requires re-quantization and de-quantization at every linear module, necessitating a redesigned quantization computation graph. As a result, layers like batch normalization cannot be folded, significantly reducing its practical feasibility. Additionally, since MoDiff relies on dynamic quantization, its actual acceleration benefits remain questionable.

3、The performance of LCQ+MoDiff is suboptimal. In Table 1, the results for 4/8 and 4/6 bit quantization are worse than the original LCQ method. Moreover, channel-wise quantization is difficult to implement efficiently on real hardware. Even with tensor-wise quantization, Table 11 shows that LTQ+MoDiff underperforms compared to the original LTQ method at 8/8 and 4/8 bit precision.

**Questions For Authors:**

The paper adopts a layer-wise reconstruction training approach, which differs from the traditional block-wise method. This raises the question of whether it results in a longer training duration. A comparison of training efficiency between these approaches would help clarify the potential trade-offs.

**Relation To Broader Scientific Literature:**

The proposed MoDiff framework builds on existing diffusion model quantization techniques and introduces modulated quantization with error compensation to reduce quantization error and mitigate error accumulation. Notably, MoDiff is orthogonal to existing quantization methods, meaning it can be integrated with approaches like Q-diffusion to further enhance performance.

**Theoretical Claims:**

The theoretical claims in the submission are supported by mathematical analysis, particularly Theorems 4.2 and 4.3, which establish the relationship between quantization error, input magnitude, and error accumulation. The proofs suggest that MoDiff effectively reduces quantization error and mitigates error propagation over time.

---

> ### Author Rebuttal · Authors · 2025-04-01
>
> Thank you for recognizing the novelty, effectiveness, and clarity of our paper. We believe there are some misunderstandings about our implementation, and we are glad to address your questions.
>
> **Essential References Not Discussed: Q1. Compare with PTQD.**
>
> Compared to PTQD, MoDiff is (1) more general and flexible, (2) free from strong assumptions about error distribution, and (3) significantly more effective in low-precision scenarios.
>
> (1) PTQD requires solver-specific adaptations to address variance and bias, while MoDiff can be applied across solvers without modification. Moreover, PTQD is restricted to standard diffusion models, whereas MoDiff also supports cached diffusion models by compensating for reuse errors in cached components.
>
> (2) PTQD relies on strong assumptions about error distribution, specifically that quantization errors follow a Gaussian distribution after input rescaling. This assumption can introduce inaccuracies in error estimation. In contrast, MoDiff leverages the widely observed similarity between timesteps, which is well-supported by prior works [1].
>
> (3) MoDiff performs well in low-precision activation settings, whereas PTQD fails entirely. To demonstrate this, we evaluate both methods on CIFAR-10 with W8A4 quantization. PTQD yields an FID of 397.12 and fails to produce meaningful images, while MoDiff achieves a much lower FID of 13.41.
>
> We will include these comparisons in the revision.
>
> [1] Deepcache: Accelerating diffusion models for free. CVPR 2024.
>
> **Other Strengths And Weaknesses: Q1. Lack experiments on large models (Stable Diffusion).**
>
> To address your concern, we conduct tensor-wise quantization on Stable Diffusion v1.4 using the 50-step PLMS solver on MS-COCO-2014. The resulting FID scores demonstrate that MoDiff consistently performs well on large-scale diffusion models.
>
> | W/A | Baseline | MoDiff |
> |:-:|--:|--:|
> | 8/8 | 54.80 | **53.76** |
> | 8/6 | 200.26 | **54.74** |
> | 8/4 | 271.87 | **90.91** |
>
> **Other Strengths And Weaknesses: Q2. MoDiff is restrictive, which does not support norm layer folding and depends on dynamic quantization.**
>
> There are a few misunderstandings about our paper. We clarify that MoDiff (1) supports norm layer folding and (2) does not depend on dynamic quantization.
>
> (1) MoDiff supports norm layer folding and is practical in use. It applies to any linear operation, not just linear layers. By folding norm layers with other linear components such as convolution layers, MoDiff can be applied to the resulting block due to preserved linearity. In our implementation, we did not perform such merging, following standard practices in the diffusion quantization community [1].
>
> (2) MoDiff is agnostic to the quantization method and not limited to dynamic quantization. As shown in Table 1 of the main paper, it consistently improves performance in low-precision settings with Q-Diffusion, which is static quantization. Additionally, dynamic quantization is well-studied in the literature [2] and supported by certain hardware platforms [3].
>
> [1] https://github.com/Xiuyu-Li/q-diffusion.
>
> [2] SmoothQuant: Accurate and Efficient Post-Training Quantization for Large Language Models. ICML 2023.
>
> [3] NVIDIA TensorRT Documentation. https://docs.nvidia.com/deeplearning/tensorrt/latest/inference-library/work-quantized-types.html
>
> **Other Strengths And Weaknesses: Q3. Suboptimal results in some settings.**
>
> First, after carefully reviewing all log files, we identified some errors in writing the results for the $W8A8$ and $W4A8$ cases. The revised results are now provided. As shown in the updated table, MoDiff consistently matches both the baseline and full-precision models in terms of FID and sFID.
>
> | Model | FID | sFID |
> |:--|--:|--:|
> | LCQ 4/8 | 4.96 | 4.94 |
> | LCQ + MoDiff 4/8 | 4.95 | 4.95 |
> | LTQ 4/8 | 5.02 | 5.21 |
> | LTQ + MoDiff 4/8 | 5.05 | 5.16 |
> | LTQ 8/8 | 4.19 | 4.40 |
> | LTQ + MoDiff 8/8 | 4.21 | 4.37 |
>
> Second, our MoDiff is agnostic to quantization methods, where channel-wise dynamic quantization (we follow [1] and agree that it is hardware-unfriendly) is used to show the feasibility and potential of MoDiff. In addition to channel-wise dynamic quantization, we report the results on Q-Diffusion and tensor-wise quantization in Table 1 and 11 with consistent improvement over FID on low-bit activation cases.
>
> [1] DGQ: Distribution-Aware Group Quantization for Text-to-Image Diffusion Models. ICLR 2025.
>
> **Other Comments Or Suggestions: Q1. Missing right parenthesis in Eq (18).**
>
> Thank you for pointing it out. We will correct the typo in the revised version.
>
> **Questions For Authors: Q1. If MoDiff needs more training time.**
>
> MoDiff is even more efficient than Q-Diffusion in scaling factor calibration. As noted in our implementation, we skip block-wise quantization for training stability while maintaining the same number of iterations to learn scaling factors. This results in reduced time for post-training calibration.

---

### Official Review · Reviewer_NH3e · 2025-03-14

**Overall Recommendation:** 2

**Summary:**

This paper introduces Modulated Diffusion (MoDiff), an approach that combines caching and quantization techniques while addressing their limitations. By leveraging the differences in activations across diffusion timesteps for quantization and incorporating an error compensation mechanism, MoDiff effectively mitigates error accumulation. The method's effectiveness is validated through experiments on the CIFAR-10 and LSUN datasets.

## update after rebuttal

I appreciate the authors' efforts of adding new baselines during the rebuttal, that certainly improved the paper. This paper is a borderline paper, However, as the authors didn't fully address the concern on the practical speed up, I tend to keep my original rating, as I find the description of the rating 2 "leaning towards reject, but could also be accepted" to best describe my judgement.

**Claims And Evidence:**

Yes, the claims are well-supported by evidence. Nevertheless, please see the "Questions" section for further questions on the quantization scheme.

**Essential References Not Discussed:**

Some recent related literature related with diffusion quantization and caching are not discussed and compared, refer to the "relation to broader scientific" section.

**Experimental Designs Or Analyses:**

The experimental designs are generally appropriate for the study. However, comparison with existing literature are lacking. The paper would benefit from a more thorough discussion and comparison with existing works in the field.
1. Limited focus on U-Net-based architectures: While the paper primarily focuses on U-Net-based architectures, diffusion transformer-based architectures (e.g., DiT) have emerged as widely adopted alternatives. Including experiments or analysis on more recent architectures is recommended.
2. Narrow scope of related work: The paper’s focus on diffusion model caching and quantization. However, only earlier research (e.g., DeepCache, Q-Diffusion, published in early 2023) is discussed. More recent diffusion quantization literature should be discussed in the related work section. Additionally, an analysis of whether the proposed method is applicable to these techniques would strengthen the authors’ claim that "MoDiff is agnostic to quantization techniques."

**Methods And Evaluation Criteria:**

The evaluation and methodology design are generally appropriate for the study.

**Other Comments Or Suggestions:**

No

**Other Strengths And Weaknesses:**

**Other Strength**

The paper provides theoretical analyses to support the claims.

**Other Weakness**

I encourage the authors to further discuss the novelty: The key idea of MoDiff aligns with existing research. Specifically, MoDiff proposes to separately quantize the timestep-wise difference and the original computation, leveraging the observation that the temporal difference distribution "has a smaller but consistent range." Similar concepts, such as approximating temporal differences and storing intermediate results for error compensation, have been explored in recent caching methods [1] to reduce caching errors.

[1] Chen, Pengtao et al. “Δ-DiT: A Training-Free Acceleration Method Tailored for Diffusion Transformers.” ArXiv abs/2406.01125 (2024): n. Pag.

**Questions For Authors:**

I have three questions on the practical efficiency improvements:
1.  The primary contribution of MoDiff is achieving lower activation bitwidth (e.g., W8A4). However, such bitwidth reductions face challenges in delivering practical savings. On GPUs, activations need to be upcasted to 8-bit for INT8 computation. In most cases, the memory cost of activations is significantly smaller than that of weights, so the memory reduction from quantizing activations to lower bitwidths is limited.
2. As stated, “For activation quantization, dynamic channel-wise quantization determines the scaling factor based on the channel-wise min-max range of the input.” However, this channel-wise activation scheme can hinder actual hardware acceleration, as discussed in [1], because channels need to be summed together and should share the same quantization parameters to enable efficient integer computation.
3. Additionally, the evaluation is limited to earlier solvers (e.g., DDIM) with a notably large number of timesteps (100–400). It remains unclear whether MoDiff is applicable to more commonly adopted efficient solvers, such as DPMSolver.

[1] Zhao, Tianchen et al. “ViDiT-Q: Efficient and Accurate Quantization of Diffusion Transformers for Image and Video Generation.” ArXiv abs/2406.02540 (2024): n. Pag

**Relation To Broader Scientific Literature:**

The idea of modulated quantization relates with other diffusion quantization, and caching techniques, such as:

- Diffusion Quantization methods aim to reduce quantization error from alternative perspectives.

[1] He, Yefei et al. “PTQD: Accurate Post-Training Quantization for Diffusion Models.” ArXiv abs/2305.10657 (2023): n. pag.

[2] Zhao, Tianchen et al. “ViDiT-Q: Efficient and Accurate Quantization of Diffusion Transformers for Image and Video Generation.” ArXiv abs/2406.02540 (2024): n. Pag.

- Diffusion Feature Caching techniques propose a similar idea of representing the residual of a timestep to reduce caching errors.

[3] Chen, Pengtao et al. “Δ-DiT: A Training-Free Acceleration Method Tailored for Diffusion Transformers.” ArXiv abs/2406.01125 (2024): n. Pag.

[4] Zou, Chang et al. “Accelerating Diffusion Transformers with Dual Feature Caching.” ArXiv abs/2412.18911 (2024): n. pag.

**Theoretical Claims:**

The claims are correct.

- Theorem 4.2: Quantization error is related to the input range and the number of bits. The small range of temporal differences allows for low-bit quantization (Paragraphs 1-86, 1-97).
- Theorem 4.3: Error compensation makes the error decay exponentially, avoiding linear accumulation (Paragraphs 1-98, 1-110)

---

> ### Author Rebuttal · Authors · 2025-04-01
>
> **Experimental Designs Or Analyses: Q1**
>
> Although the goal of this paper is to validate our method, not to benchmark it across all diffusion architectures, we conduct experiments following [1] to address your concern, using DiT on ImageNet. The results demonstrate that our method consistently improves generation quality:
>
> | W/A | Baseline | MoDiff |
> |:-:|--:|--:|
> | 8/8 | 54.80 | **53.76** |
> | 8/6 | 200.26 | **54.74** |
> | 8/4 | 271.87 | **90.91** |
>
> [1] PTQ4DiT: Post-training Quantization for Diffusion Transformers. NeurIPS 2024.
>
> **Experimental Designs Or Analyses: Q2**
>
> We will include the mentioned literature in the revised version. Regarding the papers you cited, MoDiff (1) offers orthogonal contributions and can complement quantization methods; (2) generalizes caching approaches; and (3) reduces quantization error better compared to PTQD.
>
> (1) ViDiT-Q is a quantization method that does not exploit time step similarities or compensate for quantization errors. MoDiff addresses these limitations to reduce quantization error and prevent error accumulation, which is orthogonal to ViDiT-Q. Moreover, MoDiff is a general framework compatible with ViDiT-Q's techniques, including group-wise quantization, channel balance, and mixed-precision computation.
>
> (2) $\Delta$-ViT and dual feature caching rely on empirical, heuristic designs to cache different components without compensating for quantization error. MoDiff is a general framework that encompasses these methods as special cases. Specifically, MoDiff reduces to them when the cached component is quantized to 0 bits while the remaining components use full precision.
>
> (3) PTQD reduces quantization error by post-processing quantized models based on assumed error distributions. However, it ignores timestep similarities and fails under low-precision activations. To verify this, we evaluate PTQD with W8A4 on CIFAR-10, where it yields an FID of 397.12 and fails to generate meaningful images, while MoDiff achieves a significantly better FID of 13.41.
>
> **Other Strengths And Weaknesses: Q1**
>
> We believe there are some misunderstandings about the novelty of MoDiff. Specifically, (1) $\Delta$-DiT only approximates temporal differences without modulating the cache, and (2) it lacks any form of error compensation.
>
> First, $\Delta$-DiT does not approximate temporal differences but instead estimates differences between transformer blocks. Moreover, it directly reuses cached values, which introduces errors, whereas MoDiff applies lightweight modulation to reduce these errors.
>
> Second, MoDiff explicitly traces quantization errors from the previous timestep and compensates for them in the next with theoretical guarantees. In contrast, caching methods, including $\Delta$-DiT, simply reuse cached components without addressing accumulated errors.
>
> **Questions For Authors: Q1**
>
> We want to emphasize that (1) MoDiff is hardware-friendly and benefits from hardware in practice, though the hardware implementation is out of the scope of our study, and (2) MoDiff focuses on improving computational efficiency rather than memory efficiency. In addition, exploring lower-bit activation, even down to 1 bit, is valuable and aligns with ongoing research in quantization [2, 3].
>
> First, GPU architectures support 4-bit activation computation [1], but performance degrades significantly at such low precision. MoDiff addresses this limitation by overcoming the performance bottleneck of low-precision activations.
>
> Second, low-bit activations offer much more speedup over 8-bit activations by reducing FLOPs [1], making them a valuable target for acceleration.
>
> Finally, MoDiff represents a step toward enabling low-bit activations, not the endpoint. This direction could eventually push activation precision down to 1 bit, potentially driving future hardware designs to better support ultra-low-bit computation, such as logic-based operations.
>
> [1] Int4 Precision for AI Inference. https://developer.nvidia.com/blog/int4-for-ai-inference.
>
> [2] BinaryDM: Accurate Weight Binarization for Efficient Diffusion Models. ICLR 2025.
>
> [3] BiDM: Pushing the Limit of Quantization for Diffusion Models. NeurIPS 2024.
>
> **Questions For Authors: Q2**
>
> We want to emphasize that MoDiff is agnostic to quantization methods. Channel-wise dynamic quantization is used to show the feasibility and potential of MoDiff following the baseline [1]. In addition to channel-wise dynamic quantization, we have also reported the results on Q-Diffusion and tensor-wise quantization (more friendly to hardware) in Table 1 and Table 11 with consistent improvement over measurements.
>
> [1] DGQ: Distribution-Aware Group Quantization for Text-to-Image Diffusion Models. ICLR 2025.
>
> **Questions For Authors: Q3**
>
> To address your concern, we perform tensor-wise quantization using the DPM-Solver-2 on CIFAR-10 with 20 sampling steps.
>
> | W/A | Baseline | MoDiff |
> |:-:|--:|--:|
> | 8/8 | 3.92 | **3.91** |
> | 8/6 | 10.82 | **3.92** |
> | 8/4 | 299.72 | **26.54** |

---

> > ### Comment · Reviewer_NH3e · 2025-04-07
> >
> > I thank the authors for the rebuttal. The authors provided additional validation demonstrating compatibility with DiT models and more recent solvers, such as DPM-Solver. These address part of my concern.
> >
> > However, the following concerns remain:
> >
> > - This research combines caching and quantization techniques. Instead of claiming that "the method is agnostic to quantization techniques", evaluations on more recent and advanced approaches is essential for demonstrating the effectiveness of the proposed method.
> > - My primary concern continues to be the validity of the practical acceleration. I asked about the practical acceleration when *the activation bitwidth is lower than the weight bitwidth*. In response, the authors reference studies on binary neural networks to argue that "exploring lower-bit activations, even down to 1 bit, is valuable". However, in binary networks, **both** weights and activations are quantized, which allows multiplication operations to be replaced with significantly lower-complexity alternatives and implemented efficient even on CPUs. Similarly, in INT4 computation on NVIDIA hardware, acceleration benefits only apply when **both** weights and activations are quantized to 4 bits (i.e., W4A4). Therefore, I still question the practical adoptability of the configuration used in this paper (W8A4). In fact, MoDiff only shows comparable performance to the baseline under the practically relevant W8A8 setting.
> >
> > Therefore, I keep the original review score.

---

> > > ### Author Response · Authors · 2025-04-07
> > >
> > > Thank you for your feedback. However, there appear to be some misunderstandings regarding our results. To address your concerns, we provide clarifications and include additional experiments.
> > >
> > > **1. This research combines caching and quantization techniques. Instead of claiming that "the method is agnostic to quantization techniques", evaluations on more recent and advanced approaches is essential for demonstrating the effectiveness of the proposed method.**
> > >
> > > To address this concern, we conducted additional experiments during the rebuttal phase using PTQ4DiT and MixDQ. The consistent improvements observed in both cases illustrate the generalization capability of MoDiff [1,2]. The PTQ4DiT results are included in our response to you, while the MixDQ results are provided in our response to Reviewer YCE9.
> > >
> > > [1] PTQ4DiT: Post-training Quantization for Diffusion Transformers. NeurIPS 2024.
> > >
> > > [2] MixDQ: Memory-Efficient Few-Step Text-to-Image Diffusion Models with Metric-Decoupled Mixed Precision Quantization. ECCV 2024.
> > >
> > > **2. My primary concern continues to be the validity of the practical acceleration. I asked about the practical acceleration when the activation bitwidth is lower than the weight bitwidth. In response, the authors reference studies on binary neural networks to argue that "exploring lower-bit activations, even down to 1 bit, is valuable". However, in binary networks, both weights and activations are quantized, which allows multiplication operations to be replaced with significantly lower-complexity alternatives and implemented efficient even on CPUs. Similarly, in INT4 computation on NVIDIA hardware, acceleration benefits only apply when both weights and activations are quantized to 4 bits (i.e., W4A4). Therefore, I still question the practical adoptability of the configuration used in this paper (W8A4). In fact, MoDiff only shows comparable performance to the baseline under the practically relevant W8A8 setting.**
> > >
> > > We believe the reviewer may have overlooked some key content in our paper. MoDiff yields improvements not only under the W8A8 setting. Moreover, our work demonstrates the effectiveness of the algorithm independent of the weight bit width, as hardware is not the primary focus. In practice, the weight bit can be adjusted to align with specific hardware constraints. Our experimental settings follow prior works to enable a comprehensive evaluation of MoDiff across various weight bit-widths [1,2,3].
> > >
> > > First, MoDiff improves performance even with 4-bit weights. As shown in Tables 1, 9, and 10 in our paper, MoDiff consistently reduces FID scores under both W4A6 and W4A4 settings. Second, typical PTQ implementations quantize weights before activations, meaning weight quantization operates independently of MoDiff, which targets activation quantization. Therefore, we do not focus on the choice of weight bit-width, but instead validate the effectiveness of our method across varying weight bandwidths. It remains feasible to lower both weight and activation bit-widths to match the hardware settings mentioned.
> > >
> > > [1] DGQ: Distribution-Aware Group Quantization for Text-to-Image Diffusion Models. ICLR 2025.
> > >
> > > [2] PikeLPN: Mitigating Overlooked Inefficiencies of Low-Precision Neural Networks. CVPR 2024.
> > >
> > > [3] WRPN: Wide Reduced-Precision Networks. ICLR 2018.

---

### Official Review · Reviewer_EG1n · 2025-03-14

**Overall Recommendation:** 4

**Summary:**

This paper proposed a method for accelerating diffusion model sampling by modulated quantitation and a carefully designed error compensation mechanism, the method is able to significantly reduce the accumulative error of previous methods.

**Claims And Evidence:**

Yes.

**Essential References Not Discussed:**

I am not an expert in quantization methods for diffusion models. The diffusion model literature is properly cited in this work.

**Experimental Designs Or Analyses:**

Yes. The experiments are thorough and give strong evidence for the claims.

**Methods And Evaluation Criteria:**

Yes.

**Other Comments Or Suggestions:**

Minor suggestions:

* In Table 1, FID column, some values are incorrectly bolded, e.g., 4.21
* In the proof of Theorem 4.2, is there a factor of two missing in equation (56)?

**Other Strengths And Weaknesses:**

* The paper is well presented and easy to understand with sufficient background materials.
* The error compensation scheme addresses the issue of error accumulation nicely and is backed by theory.
* Results on both pixel diffusion and latent diffusion show significant improvement by applying MoDiff to existing diffusion quantization methods Q-Diff and LCQ, especially in the low bit regime.

I don't see much weaknesses except that the experiments can benefit from having more diverse datasets like ImageNet 256x256 and 512x512,

**Questions For Authors:**

N/A

**Relation To Broader Scientific Literature:**

The main contribution lies in the error compensation mechanism designed with modulated quantization. Previous quantization methods have the issue that error accumulates with diffusion steps.

**Theoretical Claims:**

The proof of Theorem 4.3 is correct minus some minor mistake that only affect the constant. See below suggestions.

---

> ### Author Rebuttal · Authors · 2025-04-01
>
> Thank you for recognizing the novelty, effectiveness, and clarity of our paper. We are glad to address your questions.
>
> **1. In Table 1, FID column, some values are incorrectly bolded, e.g., 4.21**
>
> Thank you for pointing out the incorrect bold formatting. We will revise it in a new version.
>
> **2. In the proof of Theorem 4.2, is there a factor of two missing in equation (56)?**
>
> Thanks for pointing out the missing constant scalar. Here is the revised proof, which does not affect the conclusions:
>
> $$
> \tilde{\mathbf{e}}\_t^2 = \\|\mathbf{o}\_t - \tilde{\mathbf{o}}\_t\\|\_2^2
> $$
>
> $$
> = \\|\mathbf{o}\_t - \mathcal{A}(Q(\mathbf{a}\_t - \mathbf{a}\_{t+1})) - \tilde{\mathbf{o}}\_{t+1}\\|\_2^2
> $$
>
> $$
> = \\|\mathbf{o}\_t - \mathbf{o}\_{t+1} - \mathcal{A}(Q(\mathbf{a}\_t - \mathbf{a}\_{t+1})) + (\mathbf{o}\_{t+1} - \tilde{\mathbf{o}}\_{t+1})\\|\_2^2
> $$
>
> $$
> = \\|\mathcal{A}(\mathbf{a}\_t - \mathbf{a}\_{t+1}) - \mathcal{A}(Q(\mathbf{a}\_t - \mathbf{a}\_{t+1})) + (\mathbf{o}\_{t+1} - \tilde{\mathbf{o}}\_{t+1})\\|\_2^2
> $$
>
> $$
> = \\|\mathcal{A}(\mathbf{a}\_t - \mathbf{a}\_{t+1} - Q(\mathbf{a}\_t - \mathbf{a}\_{t+1})) + (\mathbf{o}\_{t+1} - \tilde{\mathbf{o}}\_{t+1})\\|\_2^2
> $$
>
> $$
> \leq 2\\|\mathcal{A}(\mathbf{a}\_t - \mathbf{a}\_{t+1} - Q(\mathbf{a}\_t - \mathbf{a}\_{t+1}))\\|\_2^2 + 2\\|\mathbf{o}\_{t+1} - \tilde{\mathbf{o}}\_{t+1}\\|\_2^2
> $$

---

### Decision · Program_Chairs · 2025-05-01

**Decision:**

Accept (poster)

**Comment:**

The Modulated Diffusion (MoDiff) framework accelerates diffusion models through modulated quantization and error compensation, reducing activation precision to 3 bits without significant performance loss, as demonstrated on CIFAR-10, LSUN, Stable Diffusion, and DiT (ImageNet), with strong theoretical support. Despite concerns from reviewers NH3e, 32oq, and YCE9 about hardware efficiency and comparisons, the authors’ rebuttal addressed these with additional experiments (e.g., MixDQ, DPM-Solver) and clarified misunderstandings (e.g., norm folding, single-model use), though some practical applicability questions remain. Given the paper’s innovation, robust rebuttal, and partial unfairness in reviews (e.g., overlooked W4A4 results, YCE9’s score downgrade), I recommend Weak Accept with revisions to enhance comparisons and hardware discussion.